# A plant reovirus hijacks endoplasmic reticulum-associated degradation machinery to promote efficient viral transmission by its planthopper vector under high temperature conditions

Xiangzhen Yu[☯], Dongsheng Jia[☯], Zhen Wang, Guangjun Li, Manni Chen, Qifu Liang, Yanyan Zhou, Huan Liu[iD], Mi Xiao, Siting Li, Qian Chen, Hongyan Chen*, Taiyun Wei[iD]*

Fujian Province Key Laboratory of Plant Virology, Vector-borne Virus Research Center, State Key Laboratory of Ecological Pest Control for Fujian and Taiwan Crops, Fujian Agriculture and Forestry University, Fuzhou, Fujian, PR China

☯ These authors contributed equally to this work.
* chy9948@163.com (HC); weitaiyun@fafu.edu.cn (TW)

**Data Availability Statement:** All relevant data are within the manuscript and its Supporting Information files.

## Abstract

In the field, many insect-borne crop viral diseases are more suitable for maintenance and spread in hot-temperature areas, but the mechanism remains poorly understood. The epidemic of a planthopper (*Sogatella furcifera*)-transmitted rice reovirus (southern rice black-streaked dwarf virus, SRBSDV) is geographically restricted to southern China and northern Vietnam with year-round hot temperatures. Here, we reported that two factors of endoplasmic reticulum-associated degradation (ERAD) machinery, the heat shock protein DnaJB11 and ER membrane protein BAP31, were activated by viral infection to mediate the adaptation of *S. furcifera* to high temperatures. Infection and transmission efficiencies of SRBSDV by *S. furcifera* increased with the elevated temperatures. We observed that high temperature (35°C) was beneficial for the assembly of virus-containing tubular structures formed by nonstructural protein P7-1 of SRBSDV, which facilitates efficient viral transmission by *S. furcifera*. Both DnaJB11 and BAP31 competed to directly bind to the tubule protein P7-1 of SRBSDV; however, DnaJB11 promoted whereas BAP31 inhibited P7-1 tubule assembly at the ER membrane. Furthermore, the binding affinity of DnaJB11 with P7-1 was stronger than that of BAP31 with P7-1. We also revealed that BAP31 negatively regulated DnaJB11 expression through their direct interaction. High temperatures could significantly upregulate DnaJB11 expression but inhibit BAP31 expression, thereby strongly facilitating the assembly of abundant P7-1 tubules. Taken together, we showed that a new temperature-dependent protein quality control pathway in the ERAD machinery has evolved for strong activation of DnaJB11 for benefiting P7-1 tubules assembly to support efficient transmission of SRBSDV in high temperatures. We thus deduced that ERAD machinery has been hitchhiked by insect-borne crop viruses to enhance their transmission in tropical climates.

**Funding:** This project was supported by funds from the National Natural Science Foundation of China to HC under grant number 31772132 (http://www.nsfc.gov.cn/), the National Key R&D Program of China to HC under grant number 2016YFD0300700 (http://service.most.gov.cn/sbksdy/),the National Natural Science Foundation of China to TW under grant number 31730071 (http://www.nsfc.gov.cn/), the Natural Science Foundation of Fujian Province to DJ under grant number 2020J06015 (http://xmgl.kjt.fujian.gov.cn/loginSignout.do), the National Key R&D Program of China to QC under grant number 2017YFD0200900 (http://service.most.gov.cn/sbksdy/). The funders had no role in study design, data collection and analysis, decision to publish, or preparation of the manuscript.

**Competing interests:** The authors have declared that no competing interests exist.

## Author summary

In the field, many insect-borne crop viral diseases are more suitable for maintenance in tropical and sub-tropical climates. Generally, insect heat shock proteins would be activated to ensure the proper functioning of virus-encoded proteins in high temperatures. Here, we report that a fine-regulation mechanism has evolved for strong activation of one heat shock protein of DnaJ/Hsp40 family in endoplasmic reticulum-associated degradation (ERAD) machinery by a rice reovirus for ensuring the proper assembly of abundant virus-induced tubules to support efficient viral transmission by planthopper vectors in high temperatures. Thus, insect ERAD machinery, an important protein quality control system of cells, has been hijacked by viral pathogens to enhance their survival and transmission in tropical climates. Our findings imply the complex interplay among virus-induced tubules, insect ERAD machinery and high temperatures would provide an excellent model system for deep investigation of how hot environmental temperatures drive the spread of insect-borne viruses in the field.

## Introduction

The epidemic dynamics of vector-borne viral pathogens depend on the interplay among the virus, host, vector, and environment. Insect vectors cannot control their body temperatures, so their survival and performance depend on the ambient temperatures of their habitat. Therefore, environmental temperatures have the potential to affect the spread of viral diseases transmitted by insect vectors. In general, the efficiency of mosquito-borne virus transmission peaks at moderate temperatures but declines at both high and low temperatures [1, 2]. For example, the optimal temperatures for various viral pathogens in mosquitoes have been predicted: 25˚C for west nile virus, 29˚C and 26˚C for dengue and zika viruses [3–6]. In the field, many insect-borne crop viral diseases are more suitable for maintenance and spread in tropical and sub-tropical climates, such as the brown planthopper *Nilaparvata lugens*-transmitted rice ragged stunt virus and rice grassy stunt virus, the white-backed planthopper *Sogatella furcifera*-transmitted southern rice black-streaked dwarf virus (SRBSDV), and whitefly *Bemisia tabaci*-transmitted cassava mosaic virus [7–9]. Currently, the mechanisms underlying how temperatures affect the spread of vector-borne viral diseases remain elusive.

In general, heat shock proteins (Hsps) can confer the ability of insects to adapt to temperature changes in their environments [10–12]. The Hsps generally function as molecular chaperones that serve to ensure proper folding and translocation of nascent membrane proteins in the endoplasmic reticulum (ER) [13, 14]. Newly synthesized membrane proteins undergo a strict quality-control checkpoint, and misfolded secretory proteins are targeted across the ER membrane back to the cytosol for proteasome degradation in a process known as ER-associated degradation (ERAD) [15]. The ERAD machinery is quite complex and requires the synchronization of many different proteins, such as DnaJB11/ERdj3, B-cell receptor associated protein 31 (BAP31), Derlin-1, Derlin-2, and ER Hsp70 BiP [16–22]. DnaJB11 is an ER-targeted Hsp40 chaperone required for the proper folding, assembly, trafficking and delivering of membrane proteins to the ER Hsp70 BiP for ATP-dependent chaperoning in the ERAD machinery [19–21, 23]. BAP31 is a ubiquitous ER membrane protein that has been implicated in the ER retention of transmembrane proteins, such as CFTR, cellubrevin, and cytochrome P450 2C2 [18, 24]. Many viruses such as influenza A virus, dengue virus, hepatitis C virus, tomato bushy stunt virus, tobacco mosaic virus, tomato yellow leaf curl virus, and tomato

spotted wilt virus (TSWV) have been reported to hijack host Hsps and their co-chaperones to assist in the synthesis, localizations and folding of viral proteins, facilitating viral replication, assembly and movement in infected cells [10, 25–33]. Previously, BAP31, DnaJB11 and BiP in the ERAD machinery have been shown to be essential for the dislocation of simian virus 40 from the ER to the cytosol [30, 34]. Furthermore, viral infection would increase the high temperature tolerance of insect vectors or plant hosts by stimulating the expression of Hsps [10, 12, 35, 36]. Hsps greatly contribute to thermotolerance as they act to appropriately refold/stabilize and protect proteins from high temperature inactivation [37, 38]. However, the mechanisms by which viruses activate the Hsps and their co-chaperones in the ERAD machinery of insect vectors to adapt the high temperature stress remain elusive.

SRBSDV, a plant reovirus, can replicate in and be transmitted by the white-backed planthopper (*S. furcifera*), a long-distance migratory pest of rice, and is geographically restricted to the tropical and subtropical areas throughout southern China and northern Vietnam [39, 40]. In China, the epidemics of SRBSDV are restricted to the southern portion of the Yangtze River [40]. Though *S. furcifera* can migrate northward to the northern portion of the Yangtze River of China, as well as Korea and Japan, but SRBSDV diseases have rarely been reported in these regions over the past ten years. Thus, SRBSDV diseases are more suitable for maintenance in hot-temperature areas. Previously, Xu et al. reported that SRBSDV infection decreased the low temperature (5˚C) tolerance but improved the high temperature (36˚C) tolerance of its vectors [12]. Thus, SRBSDV infection may enable its vectors to better survive and propagate in hot-temperature areas. This partially explains why greater SRBSDV spread occurs in hot-temperature areas; however, the importance of high temperatures for viral propagation and transmission in insect vectors remains undetermined. It has been reported that SRBSDV infection strongly elevated the heat shock responses in *S. furcifera* under heat stress, indicating that Hsps might be involved in the conferring the adaptation of SRBSDV to hot environments [12]. Previously, we have determined that the efficient spread of plant reoviruses in insect vectors was dependent on virus-containing tubules composed of virus-encoded membrane nonstructural proteins [7, 41–43]. Moreover, inhibition of vesicular transport from the ER with brefeldin A (BFA) abolished the assembly of such tubules [41, 44]. These data suggest that ER might be essential for the proper biogenesis of the tubules induced by plant reoviruses. The nonstructural membrane protein P7-1 of SRBSDV has the capacity to form homodimers or oligomers to assemble the proposed helical symmetry structure of tubules [43, 45, 46]. In this study, we observed that high temperatures promoted the propagation and transmission of SRBSDV by *S. furcifera*. Furthermore, we found that a new temperature-dependent protein quality control pathway in the ERAD machinery had evolved for strong activation of DnaJB11 for ensuring the proper assembly of virus-induced tubules to support efficient viral transmission by planthopper vectors in high temperatures.

## Results

### Effects of different temperatures on the development and viral infection of *S. furcifera*

To explore how high temperatures affected the infection and transmission of SRBSDV by *S. furcifera*, we firstly compared the eclosion and mortality rates of viruliferous or nonviruliferous *S. furcifera* reared under different temperature conditions (15, 20, 25 and 35˚C). We observed that high temperature increased while low temperature decreased the mortality rates of viruliferous or nonviruliferous *S. furcifera* (S1A–S1D Fig). Conversely, the high temperature promoted insect development, but the low temperature caused insect diapause (S1A–S1D Fig). However, under the high temperature (35˚C) for 6 days, the mortality rate of viruliferous *S.*

*furcifera* was lower than that of nonviruliferous *S. furcifera* (S1E and S1F Fig), consistent with the previous evidence that SRBSDV infection improved the tolerance of *S. furcifera* to high temperature [12].

We next investigated how high temperatures improved the propagation and transmission of SRBSDV by *S. furcifera*. SRBSDV establishes their initial infection in the insect midgut, disseminates to the hemolymph and finally into the salivary glands, from where the virus is introduced into susceptible hosts [47]. SRBSDV can employ virus-induced tubules composed of nonstructural protein P7-1 as a vehicle for viral spread in the body of *S. furcifera* [43] (Fig 1A). Furthermore, nonstructural protein P9-1 is the main component of viral inclusion, namely, viroplasm, for supporting viral replication and assembly of progeny virions [47, 48] (Fig 1A). We thus investigated the effects of different temperatures (15˚C, 20˚C, 25˚C and 35˚C) on SRBSDV infection in the midguts or salivary glands of *S. furcifera* by immunofluorescence microscopy using virus-induced tubule and viroplasm as the markers. At 6 days post-first access of insects to diseased plants (padp), immunofluorescence microscopy showed that virus-induced tubules of P7-1 (P7-1 tubules) and viroplasms of P9-1 distributed extensively from the midgut epithelium to the visceral muscles at 35˚C but remained restricted to the midgut epithelium at 15˚C (Fig 1B and 1C, S1 Table). Meanwhile, viral infection was extensively observed in the salivary glands at 35˚C but remained restricted to the midgut at 20˚C (Fig 1B and 1C, S1 Table). RT-qPCR and western blot assays confirmed that the high temperatures promoted the accumulation of viral proteins P7-1 and P9-1 in viruliferous insects (Fig 1D and 1E). More importantly, the transmission rates of SRBSDV by *S. furcifera* increased gradually with the elevated temperatures (S2 Table). At 35˚C, the transmission efficiency of one individual *S. furcifera* reached almost 89% (S2 Table). Thus, high temperatures appear to be beneficial for the infection and transmission of SRBSDV by *S. furcifera*.

## Interaction of SRBSDV P7-1 with ERAD proteins DnaJB11 and BAP31 of *S. furcifera*

We next identified which insect factors could mediate the formation or transport of P7-1 tubules of SRBSDV in *S. furcifera*. Yeast two-hybrid assay was used to screen a cDNA library constructed from adult *S. furcifera* to identify insect factors interacting with SRBSDV P7-1. From the library screen, 116 colonies of 207 positive ones were randomly sequenced. Of 116 sequences, 42 contigs were assembled using the SeqMan II program to eliminate duplicate clones. Finally, 26 sequences were annotated using the BLASTX program in GenBank (S3 Table). Among these candidates, the full-length sequence of BAP31 and the C-terminal peptide-binding domain of DnaJB11 in the ERAD machinery of *S. furcifera* were identified (S3 Table).

BAP31 of *S. furcifera* (GenBank accession no. MH898983.1) encoded 228 amino acids with three transmembrane helices in the N-terminal region and two coiled coils in the C-terminal region, which shared 92.6% similarity with the counterpart of *N. lugens* (Figs 2A and S2). DnaJB11 of *S. furcifera* (GenBank accession no. MK799651) consisted of a conserved J-domain in the N-terminal region, followed by a Gly/Phe-rich region, and C-terminal peptide-binding domain, which shared 98.3% similarity with the counterpart of *N. lugens* (Figs 2A and S3). Yeast two-hybrid assay identified that P7-1 of SRBSDV interacted with the full-length of BAP31 and the C-terminal peptide-binding domain of DnaJB11 (DnaJB11C), but not with other domains of DnaJB11 (Figs 2B and S4A). Interestingly, BAP31 also interacted with the C-terminal peptide-binding domain of DnaJB11 (Fig 2B). P7-1 of SRBSDV contained two putative α-helical transmembrane domains (TM1 and TM2) of 18 and 17 amino acid residues [45]. Yeast two-hybrid assay further revealed that both BAP31 and DnaJB11C interacted with the

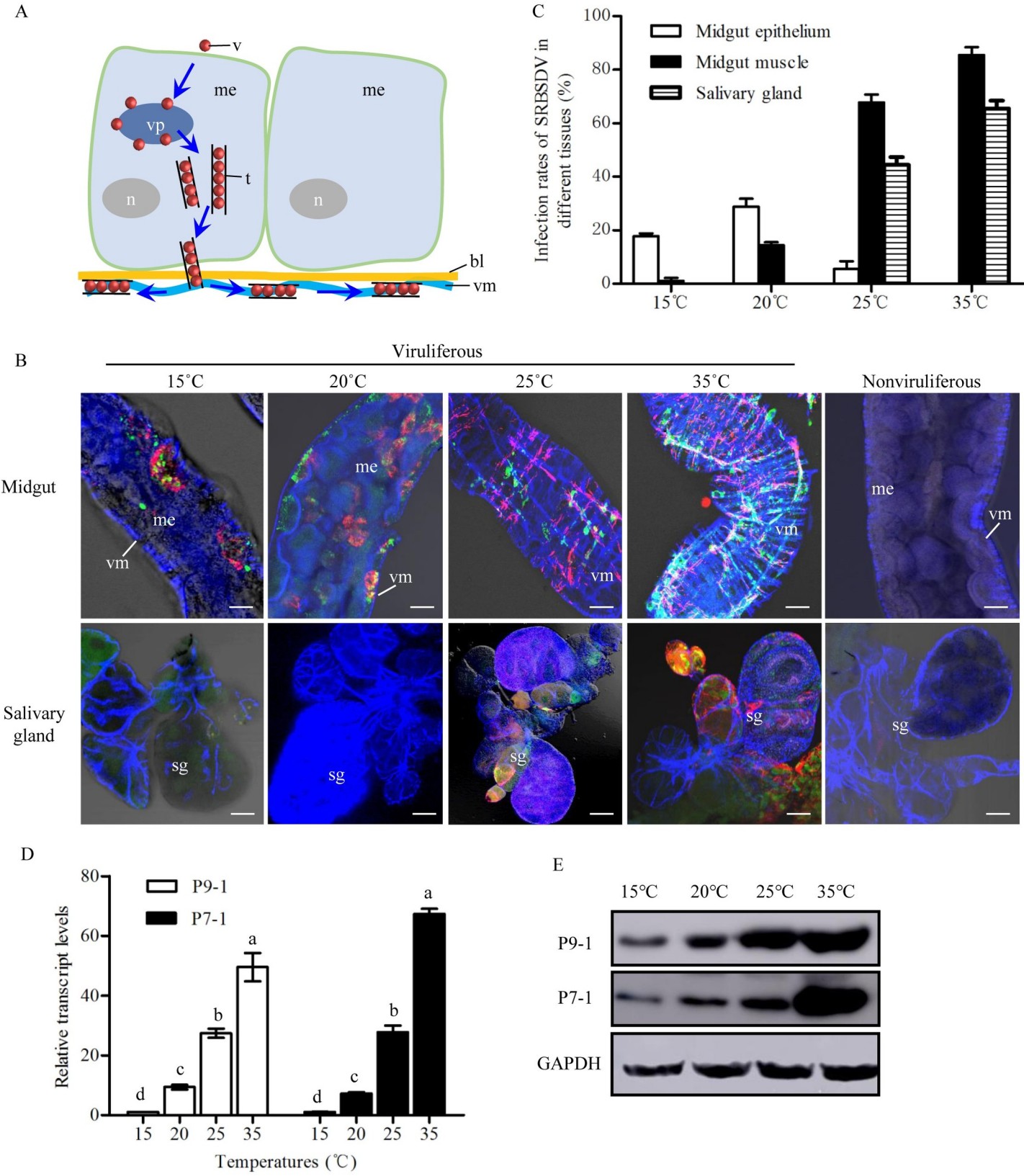

**Fig 1. Effects of temperatures on SRBSDV infection dynamics in *S. furcifera*.** (A) A model for virus-induced tubules used to overcome the midgut barriers. After virions are ingested by the insect vector, they first enter the midgut epithelium and then initiate multiplication processes for assembly of progeny virions. Some virions are packaged inside virus-induced tubules. These tubules can across the basal lamina from the epithelium into the visceral muscle to facilitate viral spread. (B) The internal organs of nonviruliferous or viruliferous *S. furcifera* under different temperatures (15, 20, 25 and 35˚C) at 6 days padp were immunolabeled with P9-1-FITC (green), P7-1-rhodamine (red), and the actin dye phalloidin-Alexa Fluor 647 (blue), then examined by immunofluorescence microscopy. Bars, 30 μm. (C) Percentages of insects positive for P7-1 and P9-1 of SRBSDV in various tissues under different temperatures (15, 20, 25 and 35˚C) at 6 days padp (n = 30), as detected by immunofluorescence microscopy. (D-E) The transcript (D) or protein (E) levels of P9-1 and P7-1 in viruliferous insects incubated at 15, 20, 25 and 35˚C at 6 days padp were measured by RT-qPCR or western blot assays. Insect GAPDH was detected with GAPDH-specific IgG as an internal control. Means (±SD) from three biological replicates are shown. Different letters in the same column indicate a significant difference (*P*<0.05, Tukey's HSD multiple test) in P7-1 or P9-1 transcript levels in viruliferous *S. furcifera* among different temperature treatments. bl, basal lamina; me, midgut epithelium; n, nucleus; sg, salivary gland; t, tubule; v, virus particle; vm, visceral muscle; vp, viroplasm.

TM1-containing N-terminus, but not with the TM2-containing C-terminus of SRBSDV P7-1 (Fig 2C). These interactions were clearly confirmed by the co-immunoprecipitation assay (Fig 2D–2G). Meanwhile, the DUAL membrane system, a variant of the yeast two-hybrid assay to study membrane protein interactions, was also used to confirm these interactions (S4B Fig).

Yeast two-hybrid assay also showed that the P7-1/DnaJB11C interaction was stronger than either the P7-1/BAP31 or BAP31/DnaJB11C interactions (Fig 2B). In competitive binding experiment by pull-down assay, we confirmed that P7-1 of SRBSDV could compete with BAP31 for binding to DnaJB11C (Fig 2H and 2I). Thus, P7-1 of SRBSDV, BAP31 and DnaJB11 formed the complex interaction relationships during viral infection of insect vectors.

The ERAD machinery contains several ER-related proteins, such as DnaJB11, BAP31, DnaJA1, DnaJA2, Derlin-1, Derlin-2, Hsp68, and BiP [16–22, 49]. However, yeast two-hybrid assay showed that P7-1 of SRBSDV did not directly interact with DnaJA1, DnaJA2, Derlin-1, Derlin-2, Hsp68, and BiP of *S. furcifera* (S4C Fig), suggesting that these ERAD components were not involved in P7-1 tubule formation during SRBSDV infection in *S. furcifera*.

## Antagonistic regulation of the assembly of SRBSDV P7-1 tubules by DnaJB11 and BAP31

The expression of the membrane protein P7-1 of SRBSDV alone induced the formation of tubules in *Spodoptera frugiperda* (Sf9) cells [45]. We initially observed P7-1 distribution in Sf9 cells infected with the recombinant baculovirus containing P7-1. A time course assay revealed that P7-1 was initially retained in the ER at 18 h post-infection (hpi), and then aggregated to form tubular structures in the cytoplasm at 36 hpi (Fig 3A and 3B). Furthermore, the treatment of BFA, the inhibitor of ER-Golgi trafficking pathway [50], abolished the assembly of such tubules and led to the redistribution of P7-1 to the ER at 36 hpi (Fig 3C and 3D). The deletion of the TMs of P7-1 abolished its ability to form tubules [45]. We further observed that the TM1-containing N-terminus but not the TM2-containing C-terminus of P7-1 was retained in the ER at 36 hpi (Fig 3E), suggesting that TM1 of P7-1 may encode the ER retention signal. Together, we showed that P7-1 was initially retained in the ER, and then secreted into the cytosol to assemble the tubules.

Expecting to confirm our suspicion that BAP31 and DnaJB11 might be involved in P7-1 tubule biogenesis, we next co-infected Sf9 cells with recombinant baculoviruses containing BAP31, DnaJB11C, or P7-1 to examine their localizations by immunofluorescence microscopy. When expressed alone, BAP31 and DnaJB11C appeared to be associated with the ER (Fig 3F). The coinfection confirmed that BAP31 and DnaJB11C were colocalized with the ER-tracker marker (Fig 3G). Whether at 18 or 48 hpi, the coinfection showed that P7-1 was always retained in the ER-like structure of BAP31 (Fig 3H). However, whether at 18 or 48 hpi, the coinfection showed that P7-1 and DnaJB11C were colocalized in the tubules (Fig 3I). Immunoelectron microscopy confirmed that DnaJB11 antibodies specifically reacted with the

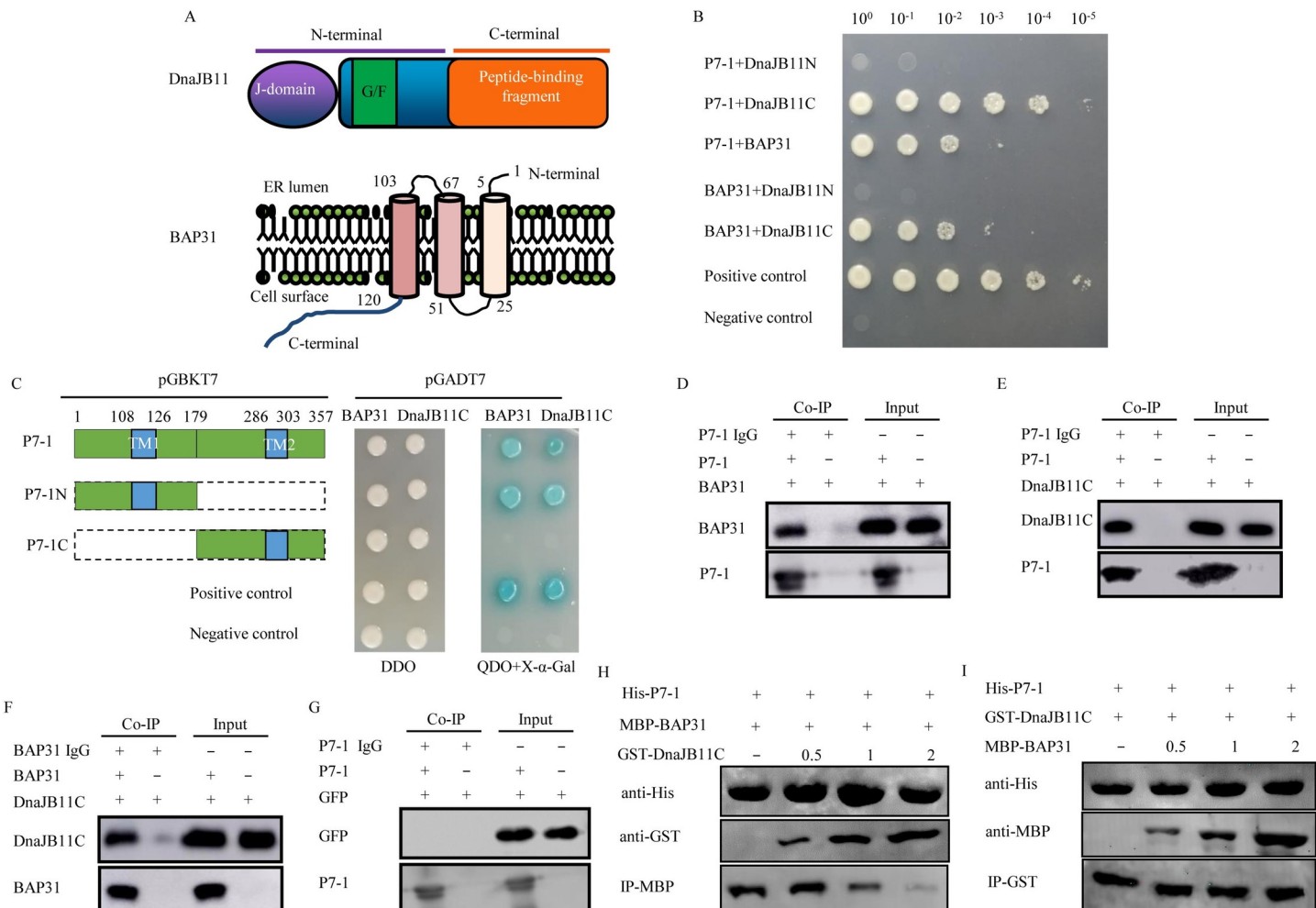

**Fig 2. Interactions among P7-1 of SRBSDV, BAP31 and DnaJB11.** (A) Structural features and domains of DnaJB11 and BAP31 from *S. furcifera*. (B) Interactions among P7-1 of SRBSDV, BAP31 and DnaJB11 were detected by yeast two-hybrid assay. Transformants on plate of QDO culture medium were labeled as follows: P7-1 +DnaJB11N, pGBKT7-P7-1/pGADT7-DnaJB11N; P7-1+DnaJB11C, pGBKT7-P7-1/pGADT7-DnaJB11C; P7-1+BAP31, pGBKT7-P7-1/pGADT7-BAP31; BAP31 +DnaJB11N, pGBKT7-BAP31/pGADT7-DnaJB11N; BAP31+DnaJB11C, pGBKT7-BAP31/pGADT7-DnaJB11C; Positive control, pGBKT7-53/pGADT7-T; Negative control, pGBKT7-Lam/pGADT7-T. Serially diluted yeast cultures were shown. (C) Both DnaJB11C and BAP31 interacted with the TM1-containing N-terminus, but not with the TM2-containing C-terminus of SRBSDV P7-1, as detected by yeast two-hybrid assay. Transformants on plate of DDO or QDO+X-α-Gal culture medium. (D-G) The interactions among P7-1, BAP31 and DnaJB11C were demonstrated by Co-IP assay. (D) The lysate was immunoprecipitated with P7-1 antibodies, and the immunoprecipitated proteins were detected by BAP31 antibodies. (E) The lysate was immunoprecipitated with P7-1 antibodies, and the immunoprecipitated proteins were detected by DnaJB11 antibodies. (F) The lysate was immunoprecipitated with BAP31 antibodies, and the immunoprecipitated proteins were detected by DnaJB11 antibodies. (G) The lysate was immunoprecipitated with P7-1 antibodies, and the immunoprecipitated proteins were detected by GFP antibodies. (H, I) The competitive interactions among P7-1 of SRBSDV, BAP31 and DnaJB11C were detected by pull-down assay. P7-1-His and MBP-BAP31 were incubated with Ni-NTA agarose beads, then GST-DnaJB11C was added to the beads; when increased the amounts of DnaJB11C, the binding between P7-1 and BAP31 was decreased (H). P7-1-His and GST-DnaJB11C were incubated with Ni-NTA agarose beads, then MBP-BAP31 was added to the beads; when increased the amounts of MBP-BAP31, the binding between P7-1 and DnaJB11C was not affected (I).

tubular structures of P7-1 (Fig 3J). Thus, the overexpression of BAP31 retained P7-1 in the ER, while the overexpression of DnaJB11C mediated the export of P7-1 from the ER for assembly of tubules, suggesting that DnaJB11 promoted while BAP31 inhibited the assembly of P7-1 tubules, respectively. Interestingly, at 48 hpi, the triply-infection showed that DnaJB11C and P7-1 were colocalized in the tubules, while BAP31 still was retained in the ER (Fig 3K), suggesting that P7-1 may be released from BAP31-P7-1 complex in the ER and then delivered to DnaJB11-P7-1 complex during the biogenesis of P7-1 tubules.

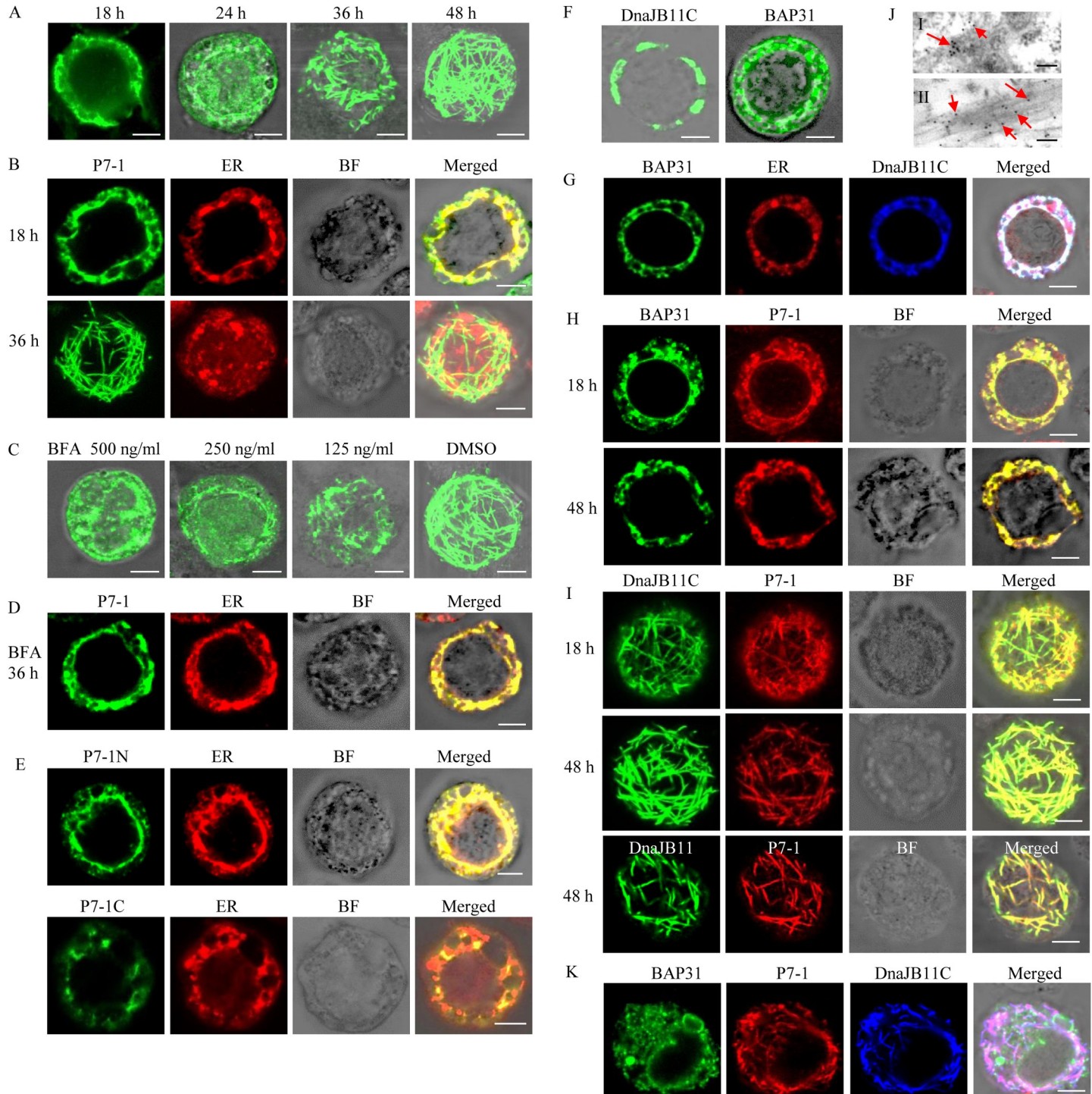

**Fig 3. Interactions among SRBSDV P7-1, DnaJB11 and BAP31 in Sf9 cells.** (A) Sf9 cells were fixed at 18, 24, 36, or 48 hpi and immunolabeled with P7-1-FITC (green). (B) After staining with the ER-Tracker Dyes (red) for 30 min, Sf9 cells were fixed at 18 or 36 hpi and immunolabeled with P7-1-FITC (green). (C) After the incubation with 500, 250, or 125 ng/mL BFA or DMSO, Sf9 cells were fixed at 36 hpi and immunolabeled with P7-1-FITC (green). (D) After staining with the ER-Tracker Dyes (red) for 30 min, 500 ng/mL BFA-treated Sf9 cells were fixed at 36 hpi and immunolabeled with P7-1-FITC (green). (E) After staining with the ER-Tracker Dyes (red) for 30 min, Sf9 cells expressed with P7-1N or P7-1C were fixed at 36 hpi and immunolabeled with P7-1-FITC (green). (F) At 48 hpi. Sf9 cells expressed with BAP31 or DnaJB11C were fixed and immunolabeled with BAP31-FITC (green) or DnaJB11-FITC (green), respectively. (G) At 48 hpi, Sf9 cells co-expressed with BAP31 and DnaJB11C were stained with the ER-Tracker Dyes (red), and then fixed and immunolabeled with BAP31-FITC (green) and DnaJB11-specifc IgG conjugated to Alexa Fluor 647 (DnaJB11-Alexa Fluor 647, blue). (H) At 18 or 48 hpi, Sf9 cells co-expressed with BAP31 and P7-1 were fixed and immunolabeled with BAP31-FITC (green) and P7-1-rhodamine (red). (I) At 18 or 48 hpi, Sf9 cells co-expressed with P7-1 and DnaJB11C or the complete DnaJB11 were fixed and immunolabeled with P7-1-FITC

(green) and DnaJB11-specific IgG conjugated to rhodamine (DnaJB11-rhodamine, red). (J) Immunogold labeling of DnaJB11 on P7-1 tubules in Sf9 cells co-expressed with P7-1 and DnaJB11C. Cells expressed with DnaJB11 alone (I) or co-expressed with P7-1 and DnaJB11C (II) were immunolabeled with DnaJB11 antibodies and goat antibodies against rabbit IgG that had been conjugated with 10-nm-diameter gold particles (arrows) as secondary antibodies. (K) At 48 hpi, Sf9 cells triply-expressed with P7-1, BAP31 and DnaJB11C were fixed and immunolabeled with BAP31-FITC (green), P7-1-rhodamine (red) and DnaJB11-Alexa Fluor 647 (blue). BF, brightfield. Bars in A-I, K, 5 μm. Bar in J, 100 nm.

We then investigated whether the knockdown of the expression of BAP31 or DnaJB11 from Sf9 cells (Sf-BAP31 and Sf-DnaJB11) by RNA interference (RNAi) could affect P7-1 tubule formation. The synthesized dsRNAs targeting Sf-BAP31 or Sf-DnaJB11 genes (dsSf-BAP31 or dsSf-DnaJB11) were transfected in Sf9 cells that have been infected with recombinant baculovirus expressing P7-1 of SRBSDV. At 36 hpi, the transcript level of Sf-DnaJB11 was significantly increased after the treatment with dsSf-BAP31, while the transcript level of Sf-BAP31 was not affected after the treatment with dsSf-DnaJB11 (Fig 4A and 4B). At 18 hpi, in dsSf-BAP31-treated Sf9 cells, P7-1 had formed abundant tubules (Fig 4C). However, at 36 hpi, in dsSf-DnaJB11-treated Sf9 cells, the P7-1 still was retained in the ER (Fig 4C). Thus, we determined that the inhibition of Sf-BAP31 expression upregulated Sf-DnaJB11 expression and in turn facilitated the P7-1 tubule assembly.

## Viral infection upregulates DnaJB11 expression but inhibits BAP31 expression to benefit P7-1 tubule assembly

RT-qPCR and western blot assays showed that DnaJB11 expression was significantly increased, while BAP31 expression was significantly decreased at the mRNA and protein levels during viral infection in *S. furcifera* (Fig 5A and 5B). At 6 days padp, in virus-infected midguts of *S. furcifera*, immunofluorescence microscopy showed that the colocalization of DnaJB11 with P7-1 tubules was extensively observed; however, only low specific staining by DnaJB11 antibodies in nonviruliferous controls was observed (Fig 5C). Immunoelectron microscopy further confirmed that DnaJB11 antibodies specifically reacted with the virus-containing tubules in virus-infected midgut of *S. furcifera* (Fig 5D). Expectedly, only low specific staining in the cytoplasm was observed by BAP31 antibodies in nonviruliferous or viruliferous insects (Fig 5E). All these results suggested that DnaJB11 expression was activated to support P7-1 tubules assembly during viral infection in insect vectors.

We then performed RNAi assay to explore the roles of DnaJB11 or BAP31 during viral infection in insect vectors. RT-qPCR and western blot assays confirmed that the accumulation of BAP31 or DnaJB11 at the mRNA or protein levels were significantly reduced by treatment with the synthesized dsRNAs targeting BAP31 or DnaJB11 genes (dsBAP31 or dsDnaJB11) in viruliferous insects, respectively (Fig 6A–6E). Interestingly, the accumulation of DnaJB11 was significantly increased after knockdown of BAP31 expression (Fig 6B and 6E), consistent with the *in vitro* results (Fig 4A). Conversely, BAP31 expression was not affected by dsDnaJB11 treatment in viruliferous insects (Fig 6A and 6E). Thus, the data suggested that BAP31 negatively regulated the expression of DnaJB11. Furthermore, RT-qPCR and western blot assays revealed that the accumulation levels of viral proteins P7-1 and P9-1 were significantly decreased in dsDnaJB11-treated insects but were significantly increased in dsBAP31-treated insects (Fig 6C–6E). Taken together, these data suggested that BAP31 and DnaJB11 exerted the antagonistic effects on SRBSDV propagation in *S. furcifera*. Immunofluorescence microscopy showed that the P7-1 tubules were extensively distributed throughout midgut visceral muscles in dsBAP31-treated insects but were restricted in the limited midgut regions in dsDnaJB11-treated insects (Fig 6F). Thus, the knockdown of DnaJB11 expression did not affect BAP31 expression but did inhibit P7-1 tubule formation. By contrast, the knockdown of

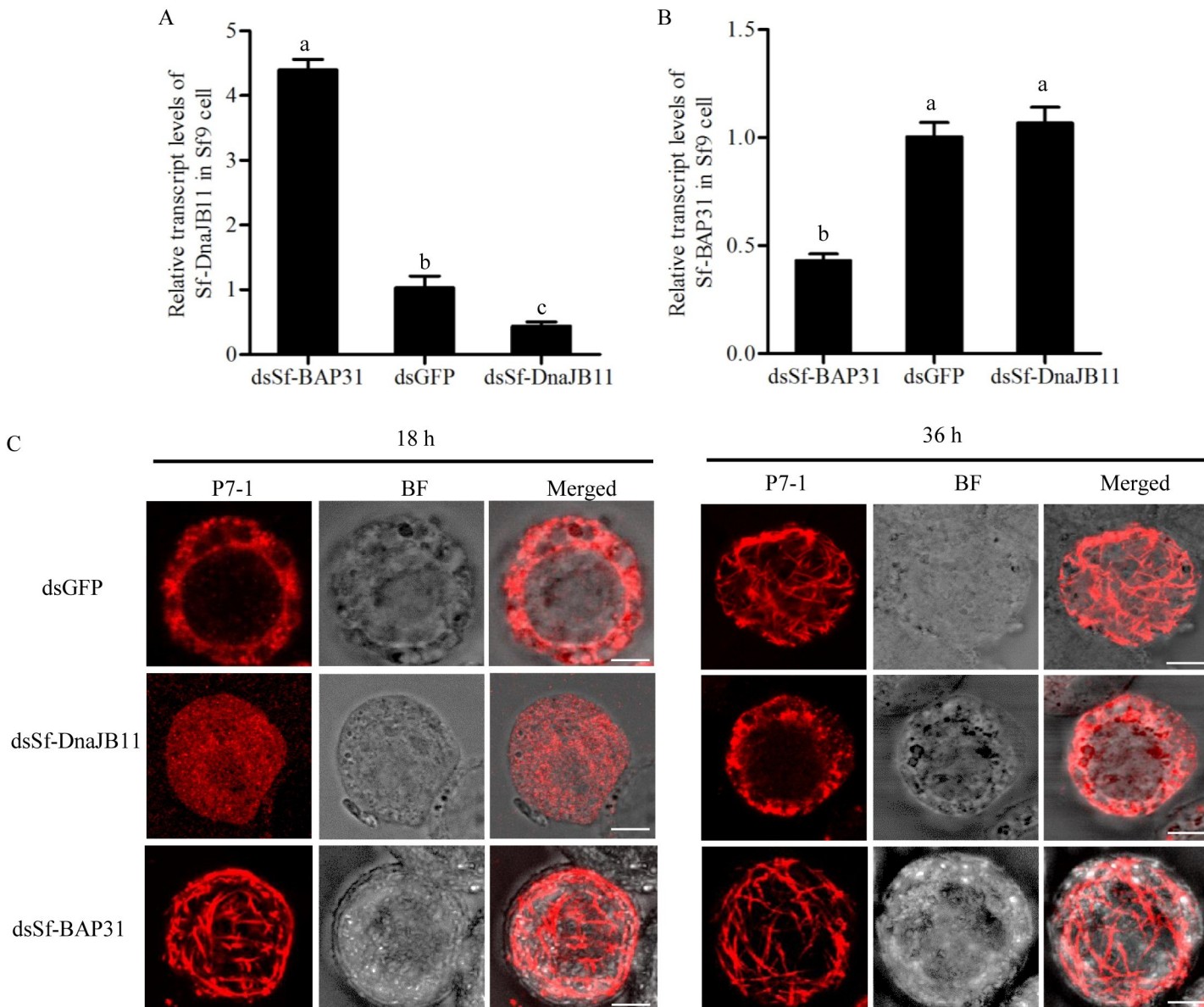

**Fig 4. Antagonistic regulation of the assembly of SRBSDV P7-1 tubules by Sf-DnaJB11 and Sf-BAP31 from Sf9 cells.** (A-B) The transcript levels of Sf-DnaJB11 (A) and Sf-BAP31 (B) in Sf9 cells infected with recombinant baculovirus containing P7-1 after treatment with dsGFP, dsSf-BAP31 or dsSf-DnaJB11 were detected by RT-qPCR assay. Means (±SD) from three biological replicates are shown. Different letters in the same column indicate a significant difference ($P < 0.05$, Tukey's HSD multiple test) in Sf-BAP31 or Sf-DnaJB11 transcript levels among *S. furcifera* treated with different dsRNAs. (C) Sf9 cells infected with recombinant baculovirus containing P7-1 after treatment with dsGFP, dsSf-BAP31 or dsSf-DnaJB11 were fixed at 18 or 36 hpi and immunolabeled with P7-1-FITC (green). BF, brightfield. Bars, 5 μm.

BAP31 expression promoted DnaJB11 expression and P7-1 tubule formation in *S. furcifera*. Accordingly, the knockdown of DnaJB11 expression significantly decreased, while the knockdown of BAP31 expression significantly increased viral transmission rates by *S. furcifera* into rice plants (Fig 6G). These results suggested that a fine-tuned regulation of BAP31 and DnaJB11 expression is beneficial for the formation of P7-1 tubules, finally facilitating viral transmission by *S. furcifera*.

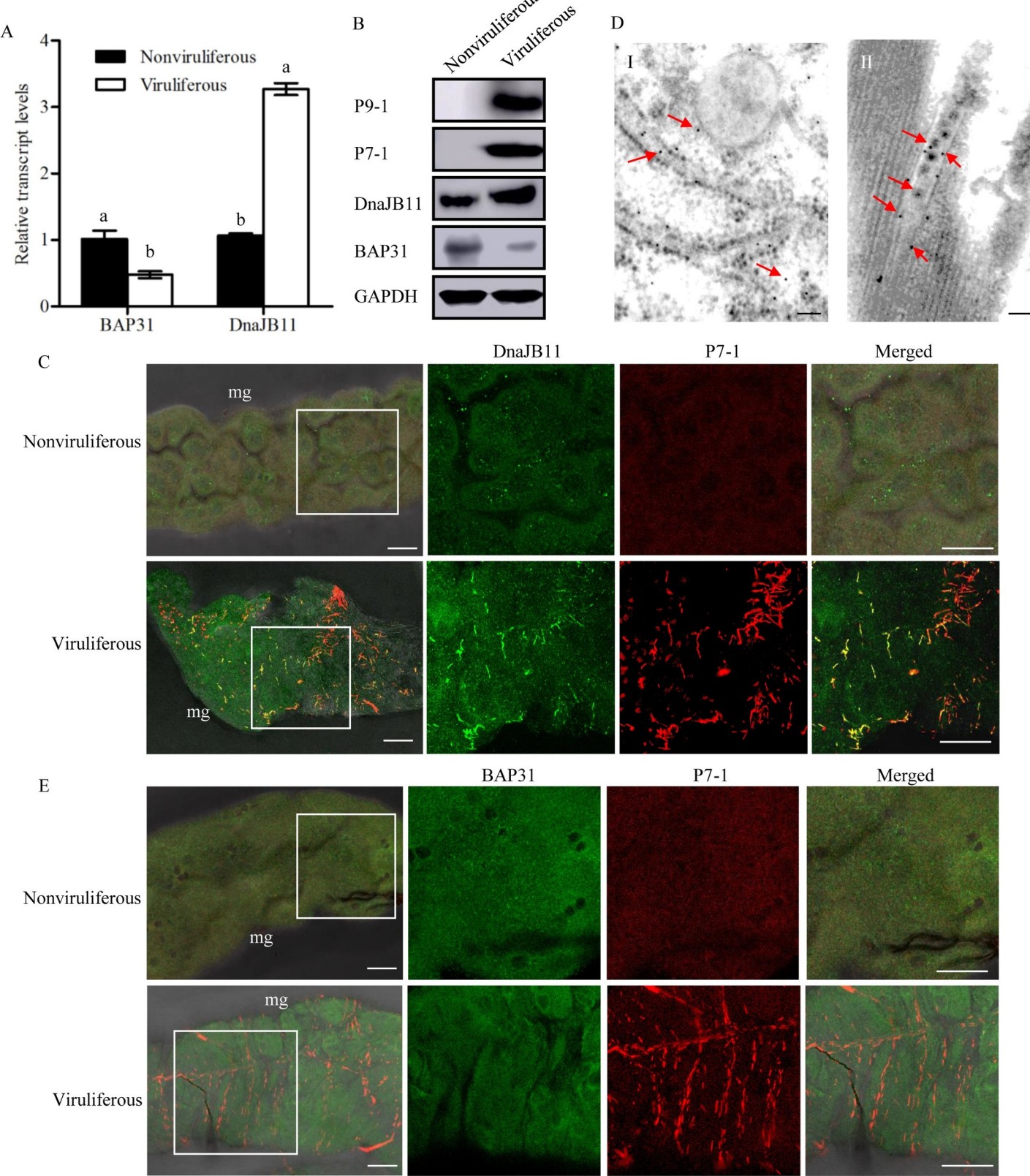

**Fig 5. Effects of SRBSDV infection on BAP31 and DnaJB11 expression in *S. furcifera*.** (A) Effects of SRBSDV infection on the transcript levels of BAP31 and DnaJB11 in viruliferous or nonviruliferous insects, as detected by RT-qPCR assay. Error bars indicate standard deviations from three biological replicates. Means (±SD) from three biological replicates are shown. Different letters in the same column indicate a significant difference ($P<0.05$, Tukey's HSD multiple test) in BAP31 or DnaJB11 transcript levels between nonviruliferous and viruliferous *S. furcifera*. (B) Effects of SRBSDV infection on the protein levels of BAP31, DnaJB11, P9-1 or P7-1 in viruliferous or nonviruliferous insects, as detected by western blot assay by using BAP31-, DnaJB11-, P9-1- or P7-1-specific IgGs. Insect GAPDH was used as an internal control. (C) At 6 days padp, the midguts of nonviruliferous or viruliferous insects were immunolabeled with DnaJB11-FITC (green) and P7-1-rhodamine (red) and then examined by immunofluorescence microscopy. DnaJB11 and P7-1 tubules were co-localized in the midgut. mg, midgut. Bars, 30 μm. (D) Immunogold labeling of DnaJB11 on virus-containing tubules in SRBSDV-infected midgut from nonviruliferous (I) and viruliferous (II) insects. The midguts were immunolabled with DnaJB11 antibodies and goat antibodies against rabbit IgG that had been conjugated with 10-nm-diameter gold particles (arrows) as secondary antibodies. Bars, 100 nm. (E) At 6 days padp, the midguts of nonviruliferous or viruliferous insects were immunolabeled with BAP31-FITC (green) and P7-1-rhodamine (red) and then examined by immunofluorescence microscopy. Bars, 30 μm.

## High temperatures significantly upregulate DnaJB11 expression but inhibit BAP31 expression

We then tested whether the interactions among SRBSDV P7-1, BAP31 and DnaJB11 were involved in regulation of SRBSDV adaptation to high temperatures. First, the expression of BAP31 and DnaJB11 in *S. furcifera* was assessed at four different temperatures (15˚C, 20˚C, 25˚C and 35˚C). In nonviruliferous and viruliferous *S. furcifera*, the transcript levels of BAP31 expression gradually decreased with the elevated temperatures; however, the expression of BAP31 was always lower in viruliferous than in nonviruliferous insects in all temperature treatments (Fig 7A). In contrast, DnaJB11 expression levels gradually increased with the elevated temperatures, and were always higher in viruliferous insects than in nonviruliferous insects in all temperature treatments (Fig 7B). In viruliferous *S. furcifera*, the levels of protein accumulation of BAP31 gradually decreased while that of DnaJB11 gradually increased with the elevated temperatures (Fig 7C). Based on these observations, it was clear that the increasing temperatures significantly promoted viral propagation and DnaJB11 accumulation but inhibited BAP31 accumulation. Taken together, it can be concluded that high temperatures can significantly upregulate DnaJB11 expression but inhibit BAP31 expression to benefit the assembly of SRBSDV P7-1 tubules for efficient viral transmission by *S. furcifera* (Fig 8).

## Discussion

The outbreak of *S. furcifera*-transmitted SRBSDV is restricted to the south of Yangtze River of China and the northern Vietnam, where the environment temperatures often stay at 36˚C in the summer. Many factors such as rice cultivars and cropping system may impact the outbreak of vector *S. furcifera* and transmission of SRBSDV. The double-cropping system for susceptible hybrid rice cultivars in the southern China are favorable for SRBSDV and *S. furcifera* occurrence [40]. Here, we determine that the propagation and transmission capability of SRBSDV by *S. furcifera* is more efficient at the high temperature (35˚C) and then declines at the lower temperatures. Though the high temperature of 35˚C tested in the laboratory is harmful to the overall performance of *S. furcifera*, it is still a suitable condition for the survival and reproduction of *S. furcifera* in summer in the southern China [51]. Furthermore, it appears that SRBSDV infection improves the tolerance of *S. furcifera* to high temperature (36˚C) [12]. Together, the environment temperatures in the southern China in the summer seasons would enable the viruliferous *S. furcifera* to efficiently survive and transmit SRBSDV, thereby aggravating the epidemic of rice viral disease. Efficient passage of viruses through different organs of insect vectors requires specific interaction between viruses and vector components [7, 52]. High temperature is beneficial for the assembly of P7-1 tubules of SRBSDV to overcome the midgut barriers encountered by *S. furcifera*, facilitating highly efficient viral transmission [43]. Here, we determine that the two ERAD machinery factors, the ER membrane protein BAP31

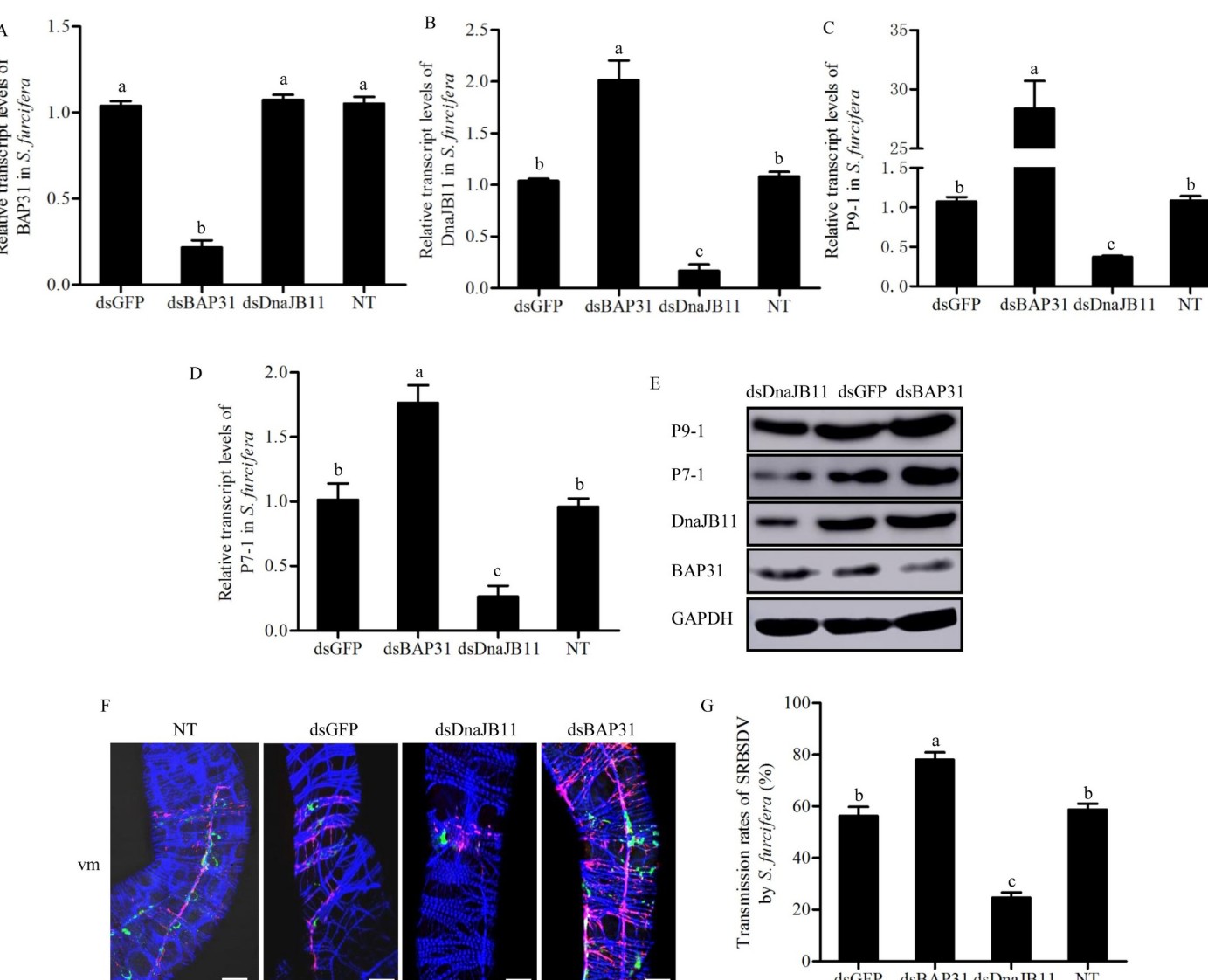

**Fig 6. SRBSDV infection upregulates DnaJB11 expression but inhibits BAP31 expression to benefit P7-1 tubule assembly and viral transmission by *S. furcifera*.**
(A-D) The transcript levels of BAP31 (A), DnaJB11 (B), P9-1 (C), or P7-1 (D) from dsRNAs (dsGFP, dsBAP31 and dsDnaJB11)-treated or untreated viruliferous insects were detected by RT-qPCR assay. Means (±SD) from three biological replicates are shown. Different letters in the same column indicate a significant difference ($P<0.05$, Tukey's HSD multiple test) in BAP31, DnaJB11, P9-1 or P7-1 transcript levels among viruliferous *S. furcifera* treated with different dsRNAs. (E) The protein levels of P9-1, P7-1, DnaJB11 or BAP31 from dsRNAs-treated viruliferous insects were detected by western blot assay by using P9-1-, P7-1-, DnaJB11-, or BAP31-specific IgGs. Insect GAPDH was used as an internal control. (F) The visceral muscles of the midguts of viruliferous dsRNAs-treated and untreated insects were immunolabeled with P9-1-FITC (green), P7-1-rhodamine (red) and the actin dye phalloidin-Alexa Fluor 647 (blue), then examined by immunofluorescence microscopy. vm, visceral muscles. Bars, 30 μm. (G) Transmission rates of SRBSDV by individual viruliferous *S. furcifera* untreated or treated with different dsRNAs. Means (±SD) from three biological replicates are shown. Different letters in the same column indicate a significant difference ($P<0.05$, Tukey's HSD multiple test) in transmission rates of SRBSDV by different dsRNAs-treated viruliferous *S. furcifera*. NT, untreated viruliferous insects.

and the Hsp DnaJB11, play the key roles in mediating the adaptation of SRBSDV-infected viruliferous *S. furcifera* to high temperatures.

SRBSDV P7-1 is initially retained in the ER, and then secreted into the cytosol to assemble the tubules. BAP31 has been implicated in the ER retention of transmembrane proteins [18, 24, 53]. We show that BAP31 binds to the TM1-containing N-terminus of P7-1, which can mediate the direct ER retention. The overexpression of BAP31 leads to the retention of P7-1 in the ER. Thus, BAP31 could act as an ER retention "receptor" for P7-1 by forming a complex

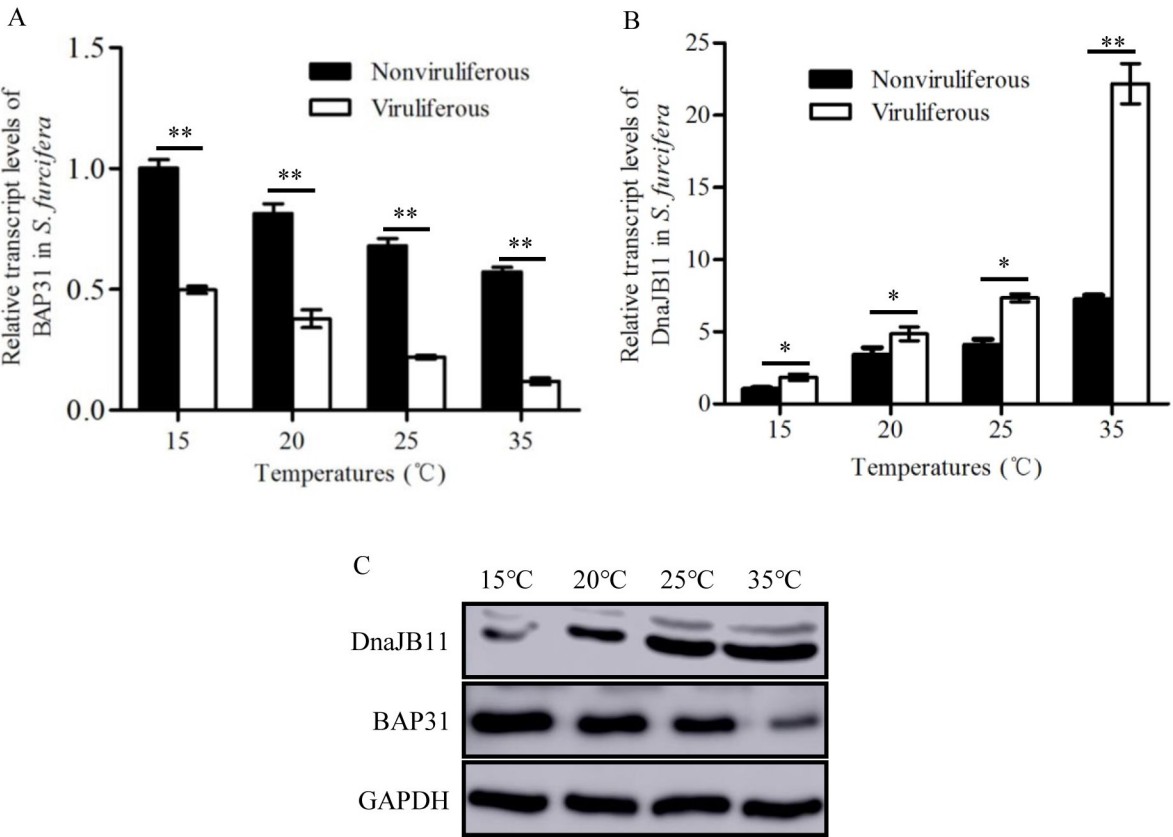

**Fig 7. Effects of temperatures on BAP31 or DnaJB11 expression in *S. furcifera*.** (A-B) The transcript levels of DnaJB11 (A) and BAP31 (B) in nonviruliferous or viruliferous insects after incubation with 15, 20, 25 and 35°C at 6 days padp were measured by RT-qPCR assay. Error bars indicate standard deviations from three biological replicates. Means (±SD) from three biological replicates are shown. Multiple comparisons of the means were conducted using a Tukey's honest significant difference (HSD) test with a one-way analysis of variance (ANOVA). *$P<0.05$, **$P<0.01$. (C) The protein levels of BAP31 or DnaJB11 in viruliferous insects after incubation with 15, 20, 25 and 35°C at 6 days padp were detected by western blot assay by using BAP31- or DnaJB11-specific IgGs. Insect GAPDH was detected with GAPDH-specific IgG as an internal control.

with P7-1. DnaJB11 competes with BAP31 for binding P7-1, and thus the overexpression of DnaJB11 leads to the delivery of P7-1 from BAP31-P7-1 complex in the ER to DnaJB11-P7-1 complex. This process finally regulates the export of P7-1 from the ER and ensures the proper assembly of tubules. We thus deduce that BAP31 as a sorting factor may just transiently retain the newly synthesized P7-1 in the ER for further folding and assembly, and then deliver it to DnaJB11-containing chaperoning complex that determine the fate of P7-1. Similarly, BAP31 has been reported to deliver the newly synthesized membrane protein CFTR to a specific ER membrane complex, the Derlin-1 complex, through the direct interaction of BAP31 and Derlin-1 [24]. SRBSDV infection inhibits BAP31 expression but upregulates DnaJB11 expression, while BAP31 negatively regulates the expression of DnaJB11 through their direct interaction. Thus, the down-regulation of BAP31 expression during viral infection in turn promotes DnaJB11 expression and tubule assembly, finally facilitating viral transmission by *S. furcifera*. Taken together, we show that a fine virus-mediated regulation of protein quality control has evolved for activation of DnaJB11 for ensuring the proper assembly of virus-induced tubules to support viral propagation and transmission by insect vectors (Fig 8).

The ERAD system contains several ER-related proteins, such as DnaJB11, BAP31, DnaJA1, DnaJA2, Derlin-1, Derlin-2, Hsp68, and ER Hsp70 chaperone BiP [16–22, 48, 53]. DnaJB11

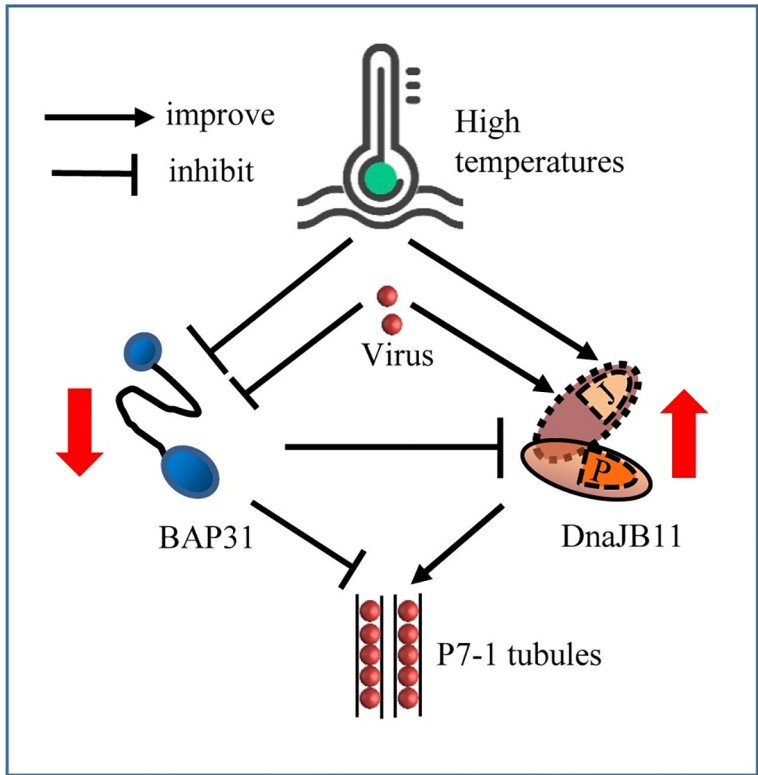

**Fig 8. Proposed model for exploitation of the ERAD machinery by SRBSDV to facilitate the proper assembly of P7-1 tubules in planthopper vector under high temperature conditions.** Two factors of ERAD machinery, DnaJB11 and BAP31 compete to interact with the tubule protein P7-1 of SRBSDV; however, DnaJB11 promotes whereas BAP31 inhibits P7-1 tubule assembly. Furthermore, BAP31 negatively regulates DnaJB11 expression through their direct interaction. Thus, high temperature can significantly upregulate DnaJB11 expression but inhibit BAP31 expression, finally facilitating the assembly of abundant P7-1 tubules.

generally binds to the membrane proteins and delivers them to BiP for ATP-dependent chaperoning; however, DnaJB11 also can binds directly to unfolded protein clients in the ER without delivering to Bip [54]. As discussed above, BAP31 binds to newly synthesized membrane proteins to ER membrane through the direct interaction of BAP31 with Derlin-1 or translocon-associated components [18, 24]. Derlin-1 is also among the hits in our yeast two hybrid screen, whereas other ERAD components are not. We have confirmed that DnaJA1, DnaJA2, Derlin-1, Derlin-2, Hsp68, and BiP of *S. furcifera* did not directly interact with SRBSDV P7-1. How the ERAD machinery manages to handle a substrate as big as the P7-1 tubules of SRBSDV remains an intriguing question that may be answered by more detailed analysis of BAP31, DnaJB11 and ERAD-associated Hsp chaperones. Furthermore, the mechanism by which the antagonistic interaction relationship between BAP31 and DnaJB11 of *S. furcifera* occurs is yet not understood. Continued studies may confirm whether such antagonistic interaction of BAP31 and DnaJB11 is a common phenomenon in nature. We believe that the newly identified protein quality control pathway via antagonistic regulation of BAP31 and DnaJB11 expression would play more important roles in ensuring the proper protein functions under different abiotic and biotic stresses.

High temperature stress further inhibits BAP31 expression but upregulates DnaJB11 expression during SRBSDV infection of *S. furcifera*, finally facilitating the assembly of more P7-1 tubules. Insect Hsps would be activated to contribute to thermotolerance as they act to

appropriately refold/stabilize and protect proteins from high temperature inactivation [10, 12, 36–38]. Our finding suggests that the high temperature stress further activates the functional role of BAP31-DnaJB11 complex in the regulation of protein quality control in viruliferous *S. furcifera*. Thus, a new temperature-dependent protein quality control pathway in the ERAD machinery has evolved for strong activation of DnaJB11 for ensuring the proper assembly of more virus-induced tubules (Fig 8), ultimately facilitating highly efficient viral transmission by insect vectors in high temperature. However, how the more ERAD-associated Hsp chaperones confer the viruliferous insects to adapt high temperatures remains an intriguing question. We deduce that insect ERAD machinery has been hitchhiked by vector-borne viral pathogens for their better maintenance and survival in hot tropical and sub-tropical climates. Our finding may be an important mechanism for explaining how global warming would drive vector-borne viral pathogens emergence, expansion and epidemic in the nature [2, 46, 55].

The nonstructural proteins of rice reoviruses such as SRBSDV P7-1, rice black-streaked dwarf virus (RBSDV) P7-1, rice dwarf virus (RDV) Pns10, and rice gall dwarf virus (RGDV) Pns11 have similar secondary structures and form the tubular structures, which facilitate viruses to overcome the tissue or membrane barriers in respective insect vectors [7, 45, 56]. Furthermore, the movement protein NSm of TSWV, a plant bunyavirus, forms the similar virus-containing tubules and interacts with plant DnaJ family proteins [25, 57]. Thus, the tubule formation induced by different plant viruses may also be mediated by Hsps through the protein quality control mechanism in the ERAD machinery. Further work will follow to determine if the virus-induced tubule formation in a temperature-dependent condition is conserved among tubule-forming plant viruses or is specific to SRBSDV-*S. furcifera* system.

## Materials and methods

### Ethics statement

Monoclonal antibodies against BAP31 was prepared in mouse by Abmart Shanghai Co. LTD, which is approved by the Science Technology Department of Shanghai Province, China with approval number SYXK (Hu) 2016–0023. Polyclonal antibodies against DnaJB11 peptide ARIRKKNEGMPVYN was prepared in rabbit by Genscript USA Innovation Company (Nanjing), which is approved by the Science Technology Department of Jiangsu Province, China with approval number SYXK (Su) 2018–0015.

### Insects, viruses, and antibodies

The nonviruliferous individuals of *S. furcifera* were collected from Fujian Province, China and propagated for several generations at $25 \pm 3°C$ in the laboratory. The SRBSDV-infected rice plants were collected from Fujian Province, China and transmitted by *S. furcifera*. Polyclonal antibodies against BAP31 of *S. furcifera* were prepared as described previously [43, 47]. Briefly, the complete open read frame (ORF) of BAP31 gene was amplified by RT-PCR and engineered into the Gateway vector pDEST17 (Thermo Fisher Scientific). The resulting plasmid pDEST17-BAP31 was transformed into *Escherichia coli* Rosetta strain and expressed by adding isopropyl-b-D-thiogalactopyranoside (IPTG). BAP31 was purified in a Ni-NTA resin column. Monoclonal antibodies against BAP31 was prepared by Abmart Shanghai Co. LTD. Polyclonal antibodies against DnaJB11 peptide ARIRKKNEGMPVYN was prepared by Genscript USA Innovation Company (Nanjing). Antibodies against P7-1, P9-1, BAP31 or DnaJB11 were conjugated directly to fluorescein isothiocyanate (FITC), rhodamine or Alexa Fluor 647 (Thermo Fisher Scientific) according to the manufacturer's instructions.

## Yeast two-hybrid assay

Yeast two-hybrid screening was performed using the Matchmaker Gal4 Two-Hybrid System3 (Clontech) according to the manufacturer's protocol. A yeast two-hybrid cDNA library with a titer of approximately $3.2 \times 10^6$ cfu/ml from total RNAs of adult *S. furcifera* was constructed in the prey plasmid pGADT7 using protocols from Clontech. The bait plasmid was constructed by cloning the full-length of SRBSDV P7-1 in the plasmid pGBKT7. To determine whether the bait fusion protein could autonomously activate the reporter genes in yeast cells, we cotransformed the bait plasmid pGBKT7-P7-1 with the positive control prey pCL1 and the empty prey plasmid pGADT7 on SD/-Ade-His (DDO), SD/-Ade-His-Leu and SD/-Ade-His-Leu-Trp (QDO) culture medium, respectively. The positive control pGBKT7-53/pGADT7-T and negative control pGBKT7-Lam/pGADT7 were cotransformed as the same way. The bait plasmid and cDNA library plasmids were transformed into yeast AH109 cells using a simultaneous co-transformation protocol. After library screening, positive clones were selected on QDO culture medium, and prey plasmids were isolated from these clones for sequencing. The sequences were used for comparison with the NCBI database using the Basic Local Alignment Search Tool (BLASTX) to obtain the annotation, length of the protein, e-value and ORF integrality, as described previously [46]. Based on the sequences of DnaJB11 (GenBank accession no. XP_022183938.1) and BAP31 (GenBank accession no. XP_022196514.1) from *N. lugens*, we obtained the complete ORFs of DnaJB11 and BAP31 genes from *S. furcifera* by homology-based cloning.

To confirm the interactions among P7-1, DnaJB11 and BAP31, the complete ORFs of P7-1 and BAP31 genes, the N-terminal (bp 1–534, P7-1N) and C-terminal (bp 535–1074, P7-1C) segments of P7-1 gene, and the N-terminal (bp 1–690, DnaJB11N) and C-terminal (bp 691–1068, DnaJB11C) segments of DnaJB11 gene were amplified and constructed in the bait plasmid pGBKT7 or the prey plasmid pGADT7, respectively. The bait and prey plasmids were used to co-transform yeast strain AH109, and β-galactosidase activity was detected on QDO +X-α-gal culture medium. The positive control pGBKT7-53/pGADT7-T and negative control pGBKT7-Lam/pGADT7-T were transformed in the same way. To detect the interaction between SRBSDV P7-1 and other ERAD components, the complete ORFs of DnaJA1, DnaJA2, Derlin-1, Derlin-2, Hsp68, and BiP genes of *S. furcifera* (the complete ORF sequences of these genes were listed in S1 Data) were amplified and constructed in the prey plasmid pGADT7. They were used to co-transform yeast strain AH109 with pGBKT7-P7-1 on DDO or QDO+X-α-gal culture medium. The negative control pGBKT7-53/pGADT7-GFP were also transformed in the same way.

Meanwhile, a DUALmembrane starter kit (DUALsystems Biotech) was used to further confirm the interactions among P7-1, BAP31 and DnaJB11C. The complete ORFs of P7-1 and BAP31 genes, and the DnaJB11N and DnaJB11C segments were constructed in the bait plasmid pBT3-STE or the prey plasmid pPR3-N. The bait and prey plasmids were co-transformed into the yeast strain NMY51 on QDO culture medium according to the manufacturer's manual. Plasmids pBT3-STE and pOST1-Nubl (positive control) or pPR3-N (negative control) were used to cotransform NMY51 in the same way.

## Competitive binding assay

The competitive binding assay was performed to detect the competitive relationship among P7-1, BAP31 and DnaJB11, as previously described [58]. Briefly, the P7-1 gene of SRBSDV was cloned into the pET-32a vector for expressing the His fusion protein (P7-1-His). The full-length ORF of BAP31 was cloned into the pMAL-c5x vector for expressing the MBP fusion protein (MBP-BAP31). DnaJB11C was cloned into the pGEX-6P-1 vector for expressing the

GST fusion protein (GST-DnaJB11C). Recombinant proteins P7-1-His, GST-DnaJB11C or MBP-BAP31 were expressed in the *E. coli* stain Rosetta and purified, respectively. The P7-1-His, MBP-BAP31 or GST-DnaJB11C were then incubated with Ni-NTA agarose beads (Qiagen) for 4 h at 4°C, and subsequently the GST-DnaJB11C or MBP-BAP31 was added in different amounts to the beads and incubated for 2 h at 4°C. After centrifuged and washed, the bead-bound proteins were detected using GST-tag, MBP-tag and His-tag antibodies by western blot assay.

## Effects of temperatures on the formation of P7-1 tubules in *S. furcifera*

To investigate the effects of SRBSDV infection in insect vectors in different temperatures, more than 400 second instar individuals of *S. furcifera* were fed on SRBSDV-infected rice plants for 2 days, and then placed on healthy rice plants at 15°C, 20°C, 25°C, or 35°C. At 6 days padp, the internal organs of 30 insects were dissected, and then fixed with 4% paraformaldehyde for 2 h and permeabilized with 0.2% Triton X-100 in 0.01 M PBS for 30 min at room temperature. The internal organs were immunolabeled with P9-1-specific IgG conjugated to FITC (P9-1-FITC), P7-1-specifc IgG conjugated to rhodamine (P7-1-rhodamine), and actin dye phalloidin-Alexa 647 (Thermo Fisher Scientific), with DnaJB11-specific IgG conjugated to FITC (DnaJB11-FITC) and P7-1-rhodamine, or with BAP31-specific IgG conjugated to FITC (BAP31-FITC) and P7-1-rhodamine. Samples were then visualized with a Leica TCS SP5 inverted confocal microscope, as previously described [59]. As controls, the nonviruliferous insects were treated exactly in the same way.

We then measured the effects of temperatures on the transcript levels of DnaJB11, BAP31, P9-1 or P7-1 genes in viruliferous or nonviruliferous insects. Total RNAs of 30 viruliferous or nonviruliferous insects in different temperatures were extracted using a Trizol Kit (Thermo Fisher Scientific) according to the manufacturer's instructions. The primer sequences were listed in S4 Table. The RT-qPCR assay was performed in triplicate using the SYBR Green PCR Master Mix kit (Promega) according to the manufacturer's instructions. The β-actin transcript of *S. furcifera* was used as the internal reference. Relative gene expression levels were calculated using the $2^{-\Delta\Delta CT}$ method [60]. Differences between samples were regarded as significant at $P <$ 0.05. Furthermore, the accumulation of DnaJB11, BAP31, P9-1 or P7-1 in viruliferous or nonviruliferous insects in different temperatures was analyzed by western blot assay with DnaJB11-, BAP31-, P9-1- or P7-1-specific IgGs, respectively.

## Effects of temperatures on development of *S. furcifera*

To determine whether temperatures affected the development of *S. furcifera*, nonviruliferous or viruliferous fourth- to fifth-instar nymphs of *S. furcifera* were reared on rice seedlings and treated with temperatures at 15°C, 20°C, 25°C, or 35°C, respectively. Each test contained three replicates and each replicate contained about 100 individuals of *S. furcifera*. The eclosion and mortality numbers were collected daily.

## Effects of temperatures on viral transmission by *S. furcifera*

To investigate the transmission rates of SRBSDV by individual *S. furcifera* in different temperatures, more than 400 second-instar nymphs of *S. furcifera* were fed on SRBSDV-infected rice plants for 2 days, and then 100 insects were placed on healthy rice plants at 15°C, 20°C, 25°C, or 35°C, respectively. At 6 days padp, the individual insects were placed in glass tubes containing a single rice seedling at a specific temperature. The insects were maintained for 10 days, with the seedlings replaced daily, as described previously [59]. Thirty insects and three replicates were conducted for the tests. The insects were collected and analyzed by RT-PCR assay at

16 days padp. The plants inoculated with the confirmed viruliferous *S. furcifera* were subjected to RT-PCR detection 10 days later. The transmission rates of SRBSDV by *S. furcifera* were calculated as the percentage of RT-PCR positive plants out of the total number of plants [61].

## Effects of knockdown of DnaJB11 or BAP31 expression by RNAi on the formation of P7-1 tubules and viral transmission by *S. furcifera*

The T7 RNA polymerase promoter was added to the forward primer and reverse primer at the 5′ and 3′ terminal to amplify a 201-bp region of DnaJB11 gene, a 387-bp region of BAP31 gene and a 714-bp region of GFP gene. PCR products were transcribed into dsRNAs *in vitro* using the T7 RiboMAX Express RNAi System, according to the manufacturer's protocol (Promega). To determine the functions of BAP31 and DnaJB11 on viral infection, 200 second-instar individuals of *S. furcifera* were microinjected with 0.5 μg/μl dsRNAs targeting the sequences from BAP31, DnaJB11 or GFP genes (dsBAP31, dsDnaJB11 or dsGFP) after feeding on SRBSDV-infected rice plants for 1 day, and then fed on healthy rice seedling for 6 days. The midguts from 30 viruliferous insects were dissected, fixed, immunolabeled with P9-1-FITC, P7-1-rhodamine, and actin dye phalloidin-Alexa 647, and then processed for immunofluorescence microscopy as described above. Furthermore, the total proteins were extracted from dsRNA-treated insects. The accumulation of DnaJB11, BAP31, P9-1 or P7-1 was analyzed by western blot assay with DnaJB11-, BAP31-, P9-1- or P7-1-specific IgGs, respectively. To measure the effects of dsRNAs on the transcript levels of DnaJB11, BAP31, P9-1 or P7-1 genes, the total RNAs were extracted from dsRNA-treated insects. Relative RT-qPCR assays were performed to analyze the relative levels of gene expression by the $2^{-\Delta\Delta CT}$ method (where CT is threshold cycle) [60]. A pool of 30 insects were used for RNA extraction and RT-qPCR assay, which was repeated three times. To investigate the transmission rates of SRBSDV by *S. furcifera* treated with dsBAP31, dsDnaJB11 or dsGFP, more than 400 second instar individuals were fed on SRBSDV-infected rice plants for 2 days, and then 100 insects were microinjected with different dsRNAs, respectively. Subsequently, the insects were placed on healthy rice seedling. At 6 days padp, the individual insects were used for viral transmission assay, as described above.

## Baculovirus expression of P7-1 of SRBSDV, BAP31 and DnaJB11 in Sf9 cells

The baculovirus system was used to express BAP31, DnaJB11, DnaJB11C, P7-1N, P7-1C, or P7-1, as described previously [47]. Recombinant bacmids were transfected into Sf9 cells using the Cellfectin II Reagent (Thermo Fisher Scientific). Sf9 cells infected with recombinant bacmids were processed for immunofluorescence microscopy. Meanwhile, Sf9 cells infected with recombinant bacmids were stained with the ER-Tracker Dyes (Thermo Fisher Scientific) for 30 min, and then processed for immunofluorescence microscopy to observe the relationship between ER and BAP31, DnaJB11C, P7-1N, P7-1C, or P7-1.

## Effects of BFA treatment or knockdown of BAP31 or DnaJB11 expression on the formation of P7-1 tubules in Sf9 cells

To observe the effects of BFA treatment on the formation of P7-1 tubules, Sf9 cells were infected with recombinant baculoviruses expressing P7-1 for 2 h, and then cultivated in the presence of 500 ng/mL, 250 ng/mL, or 125 ng/mL BFA or DMSO, as previously described [44]. After 48 hpi, the Sf9 cells were fixed with 4% paraformaldehyde and immunolabeled with P7-1-specifc IgG conjugated to FITC (P7-1-FITC), then examined by immunofluorescence microscopy. Meanwhile, Sf9 cells infected with recombinant bacmids were stained with the

ER-Tracker Dyes, and then processed for immunofluorescence microscopy to observe the relationship between ER and P7-1 of SRBSDV, as described above. To observe the effects of knockdown of Sf-DnaJB11 or Sf-BAP31 expression on the formation of P7-1 tubules, we designed primers for PCR amplification of a 656-bp segment of the Sf-BAP31 gene and a 336-bp segment of Sf-DnaJB11 gene (S5 Fig). The PCR products were used for dsRNA synthesis according to the protocol of a T7 RiboMAX Express RNAi System kit (Promega). Sf9 cells were transfected with 0.5 μg/μl dsSf-BAP31 or dsSf-DnaJB11 via the use of Cellfectin (Thermo Fisher Scientific) for 6 h and grown further in growth medium. Then, Sf9 cells were infected with recombinant baculovirus containing P7-1. At 18 and 36 hpi, the cells were fixed with 4% paraformaldehyde and immunolabeled with P7-1-FITC, then examined by immunofluorescence microscopy. Meanwhile, at 36 hpi, the total RNAs of Sf9 cells were extracted, then the effects of dsRNAs treatments on the transcript levels of Sf-BAP31 or Sf-DnaJB11 genes from Sf9 cells were measured by RT-qPCR assay, as described above.

## Co-immunoprecipitation (Co-IP) assay

Co-IP assay was performed using the Kit (Thermo Fisher Scientific) according to the manufacturer's instructions. Briefly, P7-1 antibodies were added to the resin for immobilization for 2 h at room temperature. The Sf9 cells expressing BAP31, DnaJB11C or GFP or co-expressing either BAP31/P7-1, DnaJB11C/P7-1 or GFP/P7-1 for 2 days were lysed. The supernatants were incubated with the purified P7-1-immobilized resin for 1 h at 4°C. The Co-IP fraction was eluted and collected for subsequent SDS-PAGE and western blot analysis using anti-His IgG as primary antibodies and goat anti-rabbit IgG (Sigma) as secondary antibodies. Otherwise BAP31 antibodies were added to the resin for immobilization for 2 h at room temperature. The Sf9 cells expressing BAP31 or co-expressing BAP31/DnaJB11C for 2 days were lysed. The supernatants were then incubated with the purified BAP31-immobilized resin for 1 h at 4°C. The Co-IP fraction was eluted and collected for subsequent SDS-PAGE and western blot analysis. Bands were visualized using anti-Strep IgG as primary antibodies and goat anti-mouse IgG (Sigma) as secondary antibodies.

## Immunoelectron microscopy

The midguts from viruliferous or nonviruliferous *S. furcifera* or Sf9 cells infected with recombinant baculoviruses were fixed, dehydrated, and embedded, and thin sections were cut as previously described [41]. Sections were then incubated with DnaJB11-specific IgG and immunolabeled with goat antibodies against rabbit IgG that had been conjugated with 10-nm-diameter gold particles (Sigma).

## Sequence alignment and phylogenetic analysis

Amino acid sequences of DnaJB11 and BAP31 from various species were downloaded from NCBI. Multiple sequence alignments were generated using MEGA5 [62], and the phylogenetic tree for DnaJB11 or BAP31 was generated using IQ-tree [63]. The maximum likelihood method was used to construct the phylogenetic tree with bootstrap values of 1000.

## Statistical analyses

All data were analyzed using SPSS, version 17.0. Percentage data were arcsine square-root transformed prior to analysis. Multiple comparisons of the means were conducted using a Tukey's honest significant difference (HSD) test with a one-way analysis of variance (ANOVA). The data were back-transformed after analysis for presentation in the text and figures.

## Supporting information

**S1 Fig. Effect of different temperatures on the eclosion and mortality rates of *S. furcifera*.**
(A-D) The eclosion (A, C) and mortality (B, D) rates of nonviruliferous or viruliferous *S. furcifera*. Nonviruliferous or viruliferous fourth- to fifth-instar nymphs of *S. furcifera* were treated with temperatures at 15˚C, 20˚C, 25˚C, or 35˚C for different days, and the eclosion and mortality numbers were calculated daily. Each test contained three replicates and each replicate contained about 100 individuals of *S. furcifera*. Means (±SD) from three biological replicates are shown. The statistical significance in A-D are related to the 25˚C control. $*P<0.05$. (E, F) The comparison of eclosion (E) or mortality (F) rates between nonviruliferous and viruliferous *S. furcifera* under different temperatures for 6 days. Each test contained three replicates and each replicate contained about 100 individuals of *S. furcifera*. Means (±SD) from three biological replicates are shown. $*P < 0.05$. ns, not statistically significant.
(TIF)

**S2 Fig. Sequence alignments of BAP31 from *S. furcifera* and other insect species.** (A) The amino acid homology between BAP31 from *S. furcifera* and other insect species. (B) Comparison between deduced amino acid sequences of BAP31 from *S. furcifera* and other insect species. Three predicted transmembrane helices are highlighted in blue and labeled with TM1, TM2 and TM3 in the N-terminus. The two coiled coils are labeled with CC1 and CC2 and marked with green lines in the C-terminus. (C) Phylogenetic tree of BAP31 amino acid sequences from *N. lugens*, *Laodelphax striatellus*, *Drosophila melanogaster* and *Aedes aegypti*. Numbers at each branch indicate the percentage of times a node was supported in 1000 bootstrap replicates.
(TIF)

**S3 Fig. Sequence alignments of DnaJB11 from *S. furcifera* and other insect species.** (A) The amino acid homology between DnaJB11 from *S. furcifera* and other insect species. (B) Comparison between deduced amino acid sequences of DnaJB11 from *S. furcifera* and other insect species. The conserved J-domain including Helix I region (highlighted with light purple), Helix II region (highlighted with purple), HPD motif (highlighted with yellow), Helix III region (highlighted with light blue), and Helix IV region (highlighted with blue) are marked by a red box, and the Glycine-rich regions are designated by a green box. (C) Phylogenetic tree of DnaJB11 amino acid sequences from *S. furcifera*, *N. lugens*, *L. striatellus*, *D. melanogaster* and *A. aegypti*. Numbers at each branch indicate the percentage of times a node was supported in 1000 bootstrap replicates.
(TIF)

**S4 Fig. Investigation of the interactions of SRBSDV P7-1 with different ERAD-related factors of *S. furcifera* by yeast two-hybrid assay.** (A) The autoactivation test of P7-1 bait fusion plasmid. The pGBKT7-P7-1 was cotransformed with the control plasmid pCL1 or pGADT7-T and grown on selective SD medium. Coexpression of pGBKT7-P7-1 with pCL1 resulted in reporter gene activation as shown by growth of the yeast transformants, whereas co-expression of pGBKT7-P7-1 with pGADT7-T did not yield any yeast transformant growth on selective medium. pGBKT7-53 and pGADT7-T were used as positive controls; pGBKT7-Lam and pGADT7-T were used as negative controls. (B) The interactions of SRBSDV P7-1 with BAP31, DnaJB11N, or DnaJB11C of *S. furcifera* were tested by yeast two hybrid assay using a DUAL-membrane starter kit. The transformants were plated on QDO culture medium. +, positive control (pBT3-STE/pOst1-NubI); −, negative control (pBT3-STE/pPPR3-N); P7-1+-DnaJB11C, pBT3-STE-P7-1/pPR3-N-DnaJB11C; P7-1+DnaJB11N, pBT3-STE-P7-1/pPR3-N-DnaJB11N; P7-1+BAP31, pBT3-STE-P7-1/pPR3-N-BAP31; BAP31+DnaJB11N,

pBT3-STE-BAP31/pPR3-N-DnaJB11N; BAP31+DnaJB11C, pBT3-STE-BAP31/pPR3-N-DnaJB11C. (C) The interactions of SRBSDV P7-1 with Derlin-1, Derlin-2, DnaJA1, DnaJA2, Hsp68, or BiP of *S. furcifera* were tested by yeast two-hybrid assay. Transformants were plated on DDO or QDO+X-α-Gal culture medium.
(TIF)

**S5 Fig. Sequence alignments of DnaJB11 and BAP31 between *S. furcifera* and *S. frugiperda*.** (A-B) Comparison between nucleotide sequences of DnaJB11 (A) and BAP31 (B) from *S. furcifera* and *S. frugiperda*. Nucleotide sequences of DnaJB11 and BAP31 from *S. frugiperda* shared 59% and 54% similarity with that from *S. furcifera*, respectively.
(TIF)

**S1 Table. Occurrence of P7-1 of SRBSDV in various tissues of vector at 6 days padp, as detected by immunofluorescence microscopy.**
(DOC)

**S2 Table. Transmission rates of SRBSDV by individual *S. furcifera* treated with different temperatures.**
(DOC)

**S3 Table. Putative proteins of *S. furcifera* interacted with SRBSDV P7-1 in yeast-two hybrid system.**
(DOC)

**S4 Table. Primers used in this study.**
(DOC)

**S1 Data. The complete ORF sequences of Hsp68, Derlin-1, Derlin-2, DnaJA1, DnaJA2 and BiP from *Sogatella furcifera*.**
(DOC)

## Author Contributions

**Data curation:** Dongsheng Jia, Huan Liu, Mi Xiao.

**Formal analysis:** Dongsheng Jia, Taiyun Wei.

**Funding acquisition:** Dongsheng Jia, Qian Chen, Hongyan Chen, Taiyun Wei.

**Investigation:** Xiangzhen Yu, Zhen Wang, Guangjun Li, Manni Chen, Yanyan Zhou, Huan Liu, Siting Li.

**Methodology:** Xiangzhen Yu, Dongsheng Jia, Qifu Liang.

**Project administration:** Taiyun Wei.

**Visualization:** Dongsheng Jia, Hongyan Chen.

**Writing – original draft:** Dongsheng Jia, Taiyun Wei.

**Writing – review & editing:** Dongsheng Jia, Taiyun Wei.

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
