## [Decision Letter · Decision Letter 0]

22 Oct 2020

Dear Dr. Wei,

Thank you very much for submitting your manuscript "A plant reovirus hijacks endoplasmic reticulum-associated degradation machinery to promote efficient viral transmission by its planthopper vector under high temperature conditions" for consideration at PLOS Pathogens. As with all papers reviewed by the journal, your manuscript was reviewed by members of the editorial board and by several independent reviewers. In light of the reviews (below this email), we would like to invite the resubmission of a significantly-revised version that takes into account the reviewers' comments.

We cannot make any decision about publication until we have seen the revised manuscript and your response to the reviewers' comments. Your revised manuscript is also likely to be sent to reviewers for further evaluation.

Sincerely,

Xiao-Wei Wang, PhD

Guest Editor

PLOS Pathogens

Peter Nagy

Section Editor

PLOS Pathogens

Kasturi Haldar

Editor-in-Chief

PLOS Pathogens

orcid.org/0000-0001-5065-158X

Michael Malim

Editor-in-Chief

PLOS Pathogens

orcid.org/0000-0002-7699-2064

Reviewer's Responses to Questions

**Part I - Summary**

Reviewer #1: The authors report that a plant virus SRBSDV is able to hijack the ERAD pathway in vectors by suppressing BAP31, which negatively regulates a heat shock protein (DnaJB11), and therefore ensure the proper assembly of tubules needed for viral transmission under high temperatures in vectors. They used appropriate methodologies to support their claims, such as Y2H, co-IP, as well as molecular and cellular imaging approaches. In addition, RNAi was also conducted to knock down DnaJB11 and BAP31 expression both in vitro and in vivo. The evidences are so strong that the causality of virus infection, high temperature tolerance, and DnaJB11 and BAP31 proteins are well demonstrated and clear. Although roles of BAP31 and DnaJB11 in virus transmission have been reported before, this research does deepen our understanding of temperature-mediated virus resistance.

The manuscript is in general well written and figures are correct. All reviewers’ comments are responded very well. I believe that this work is extensive and systematic and is acceptable to PLOS Pathogens.

Reviewer #2: Yu et al. report that southern rice blackstreaked dwarf virus (SRBSDV) could hijack endoplasmic reticulum-associated degradation (ERAD) machinery for ensuring the proper assembly of abundant virus-induced tubules necessary for the efficient transmission of the virus by Sogatella furcifera in the manuscript entitled with "A plant reovirus hijacks endoplasmic reticulumassociated degradation machinery to promote efficient viral transmission by its planthopper vectors under high temperature conditions". The manuscript is clearly written and contain very few spelling and grammatical errors. Some easily editable typos remain. The manuscript present a series of beautiful experiments.

However, the first section of results is titled “Infection efficiency of SRBSDV in S. furcifera increases with the elevated temperatures”. The first sentences of this section are a soup of ideas going from the viral nonstructural protein P7-1 and their role to the effect of temperature on SRBSDV infection in S. furcifera. This section doesn’t really discuss the infection efficiency but the accumulation of non-structural viral proteins. This part also describes the effect of temperature in the vector fitness. Please reorganize and/or streamline. No statistics are shown for the comparisons, therefore those cannot be assessed. Fig S1, legend describes letters for post-hocs, however images have *. Please correct. It is not clear what the * mean in regards to the comparisons, so the results and conclusions can’t be analyzed or interpreted. Further, for the effect of the temperature in insect fitness, no comparisons were performed between viruliferous and aviruliferous insects. So, sentences such as line 157 “However, under the high temperature (35˚C) for 6 or 7 days, the mortality rate of viruliferous S. furcifera was lower than that of nonviruliferous S. furcifera” are not supported by the data. Fig S1 E is supposed to represent infection rate of different organs of the vector. I do not see how the graph reports that: what organs? How many insects? Which sex, stage, etc? All this section needs serious editing.

The authors then describe the identification of protein interacting with by Y2H. There is no information about the library that was screened. Please report what was used as RNA source. There is no information as to if the library was screened by mating or co-transformation. The authors do not mention testing for autoactivation of the system with either the prey or the bait, which is a serious problem of the Y2H system. This is particular necessary since the authors identified interactions with proteins containing membrane domains. One of the drawbacks of the Matchmaker system used is that it CANNOT detect interactions with membrane proteins. I appreciate the efforts of the authors performing the validations by Co-IP, however, the identified proteins are what could be considered “sticky” which might randomly interact with proteins. The co-localizations help support the argument for interaction.

Finally, the authors perform a series of experiments to confirm the potential role of the identified proteins in the formation of the virus-induced tubules. The work is interesting but remains a correlation. I am not certain that sentences such as (line 285 ) “it can be concluded that viral infection significantly upregulated DnaJB11 expression but inhibited BAP31 expression to benefit the assembly of SRBSDV P7-1 tubules for efficient viral transmission by S. furcifera” is supported by the data. The causation (mechanism by which the virus manipulate gene expression) was not identified. The other thing that I missed was the description of the feeding behavior of the insect at different temperatures, which of course can alter acquisition and inoculation.

Reviewer #3: This is a good manuscript contributing to our understanding the transmission of plant pathogens by insect vectors. Specifically, the involvement of ER and ER-related proteins and mechanisms (like ERAD) in the transmission. The involvement of ER mechanisms in the transmission is not new and it has been shown for other systems although those two proteins are new.

**Part II – Major Issues: Key Experiments Required for Acceptance**

Reviewer #1: None.

Reviewer #2: Statistics of the experiments need to be better described and results of the analyses reported.

The Y2H system needs to have the proper controls. Test for autoactivation and I recommend testing bait and prey with unrelated proteins (can be the ones that come as controls with the kit). Same for the co-IP.

For the silencing, I did not see the expression of the untreated insects, sometimes there is an effect with the GFP control.

Reviewer #3: The current manuscript suggests an interesting hypothesis by which a plant virus, southern rice black-streaked dwarf virus, SRBSDV, which is transmitted by the planthopper Sogatella furcifera in rice uses ER machinery to promote its own transmission under high temperatures. With 35 degrees the acquisition and transmission of the virus is significantly increased. This is achieved by the overexpression of DnaJB11 and BAP31 genes upon virus acquisition by the insect. DnaJB11 facilitates the formation of viral tubules with the P7-1 nonstructural protein of the virus which in turn facilitates transmission. The relationships between the two candidate proteins studied here are antagonistic, while fine tuned interactions ensure these functions in virus transmission. BAP31 negatively regulates the expression of DnaJB11 and under high temperatures BAP31 is negatively regulated while DnaJB11 is upregulated leading to higher transmission.

The authors have performed a wealth of experiments, and I must say mostly in vivo experiments to demonstrate the relationships between P7-1, DnaJB11 and BAP31 and the formation of viral structures for transmission. Those experiments included expression at the gene and protein levels, in vitro interactions using pull down assays and Y2H system and interaction at the cell culture levels using SF9 cells. In the SF9 system, knockdown assays were used to demonstrate these relationships.

The quality of the experiments is convincing and no doubt that both proteins influence the virus dynamics in the insect as demonstrated in the immunofluorescence, interaction and knockdown experiments in both dissected organs and cell cultures.

The main drawback which I think is a must for accepting this manuscript is the ability of the authors to demonstrate the role of these interactions in the virus transmission by the insect, meaning transmission experiments following introducing interference like knockdown or specific antibodies. This is the ultimate result which should demonstrate that upon knockdown of one or both of these proteins in the insect, the transmission is affected. I am not sure whether such methods are developed for this insect-virus system, but as judged from other systems, I guess it should not be a problem. The demonstrated interaction are all in vivo interactions which might involve other proteins that influence the virus persistence in the insect, suggesting that the two identified proteins have a secondary role in the transmission. It is known from other pathogen-vector systems that the ERAD machinery influences several steps in the transmission of pathogens, and with this machinery, many proteins are involved. It cannot be ruled out that some of the observed interactions shown in this manuscript involve other factors that have not been identified here, which have a more important role than the ones identified here.

As mentioned above, several ER-related proteins function in the ERAD machinery. Although only the two proteins studied here were found to interact in the initial screen, the authors could test other ER and membrane associated proteins which could function exactly as these tow proteins. Since the system has been established and these confirmation methods have been used, other candidates could easily be screened.

The mechanisms by which the interactions between the two candidates are happening, how their antagonistic relationships are occurring are yet not understood, and the manuscript lacks this significant addition to understanding the transmission. One hypothesis might be other HSPs? Why other such proteins were not tested as it is known from other works?

Throughout the experiments controls were not included and those would be must. For example Figure 5D in which a negative control is not shown for this TEM experiment. Several other relevant controls with the immunocolocalization experiments were not included.

**Part III – Minor Issues: Editorial and Data Presentation Modifications**

Reviewer #1: Why do the authors use the fall armyworm cell lines when they have already developed the cell cultures of Sogatella furcifera (Jia et al., 2012. Journal of Virology, 86, 5800 - 5807).

Line 313, SRBSDD should be SRBSDV.

Line 319, I prefer “it appears that SRBSDV infection improves the tolerance of S. furcifera to high temperature (36˚C)”.

The authors have provided sufficient evidence that BAP3 negatively interacts with DanJB11. How do these proteins bind to each other? Would it be possible for authors to predict potential protein binding sites via bioinformatics analysis?

Reviewer #2: Rearrange first section of results, complete materials and methods.

Reviewer #3: None

PLOS authors have the option to publish the peer review history of their article (what does this mean?). If published, this will include your full peer review and any attached files.

Reviewer #1: No

Reviewer #2: No

Reviewer #3: No
---

## [Editor Report · Decision Letter 1]

11 Jan 2021

Dear Prof. Wei,

Thank you very much for submitting your manuscript "A plant reovirus hijacks endoplasmic reticulum-associated degradation machinery to promote efficient viral transmission by its planthopper vector under high temperature conditions" for consideration at PLOS Pathogens. As with all papers reviewed by the journal, your manuscript was reviewed by members of the editorial board and by several independent reviewers. The reviewers appreciated the attention to an important topic. Based on the reviews, we are likely to accept this manuscript for publication, providing that you modify the manuscript according to the review recommendations.

In the manuscript, you have emphasized that insect proteins have evolved to ensure the proper assembly of virus-induced tubules to support viral propagation and transmission by insect vectors (e.g. L44-47 in the abstract; L55-56 in the author summary; L365-368 in the discussion). However, I don’t think this is the case. In general, the insect doesn't evolve to ensure virus infection. In addition, in several places, generic conclusion was made based on studies of one virus (SRBSDV), e.g. L47-48 in the abstract; L58-60 in the author summary and L385-387, L400-410 in the discussion. The authors need to re-phrase these sentences.

Sincerely,

Xiao-Wei Wang, PhD

Guest Editor

PLOS Pathogens

Peter Nagy

Section Editor

PLOS Pathogens

Kasturi Haldar

Editor-in-Chief

PLOS Pathogens

orcid.org/0000-0001-5065-158X

Michael Malim

Editor-in-Chief

PLOS Pathogens

orcid.org/0000-0002-7699-2064

In the manuscript, you have emphasized that insect proteins have evolved to ensure the proper assembly of virus-induced tubules to support viral propagation and transmission by insect vectors (e.g. L44-47 in the abstract; L55-56 in the author summary; L365-368 in the discussion). However, I don’t think this is the case. In general, the insect doesn’t evolve to ensure virus infection. In addition, in several places, generic conclusion was made based on studies of one virus (SRBSDV), e.g. L47-48 in the abstract; L58-60 in the author summary and L385-387, L400-410 in the discussion. The authors need to re-phrase these sentences.
---

## [Editor Report · Decision Letter 2]

29 Jan 2021

Dear Dr. Wei,

We are pleased to inform you that your manuscript 'A plant reovirus hijacks endoplasmic reticulum-associated degradation machinery to promote efficient viral transmission by its planthopper vector under high temperature conditions' has been provisionally accepted for publication in PLOS Pathogens.

Best regards,

Xiao-Wei Wang, PhD

Guest Editor

PLOS Pathogens

Peter Nagy

Section Editor

PLOS Pathogens

Kasturi Haldar

Editor-in-Chief

PLOS Pathogens

orcid.org/0000-0001-5065-158X

Michael Malim

Editor-in-Chief

PLOS Pathogens

orcid.org/0000-0002-7699-2064
---

## [Editor Report · Acceptance letter]

24 Feb 2021

Dear Dr Wei,

We are delighted to inform you that your manuscript, "A plant reovirus hijacks endoplasmic reticulum-associated degradation machinery to promote efficient viral transmission by its planthopper vector under high temperature conditions," has been formally accepted for publication in PLOS Pathogens.

Best regards,

Kasturi Haldar

Editor-in-Chief

PLOS Pathogens

orcid.org/0000-0001-5065-158X

Michael Malim

Editor-in-Chief

PLOS Pathogens

orcid.org/0000-0002-7699-2064